# Soil organic carbon is a key determinant of $CH_4$ sink in global forest soils

Jaehyun Lee[1], Youmi Oh [2,3], Sang Tae Lee[4], Yeon Ok Seo[5], Jeongeun Yun [1], Yerang Yang[1], Jinhyun Kim[1,6], Qianlai Zhuang [7] & Hojeong Kang [1] ✉

Soil organic carbon (SOC) is a primary regulator of the forest–climate feedback. However, its indicative capability for the soil $CH_4$ sink is poorly understood due to the incomplete knowledge of the underlying mechanisms. Therefore, SOC is not explicitly included in the current model estimation of the global forest $CH_4$ sink. Here, using in-situ observations, global meta-analysis, and process-based modeling, we provide evidence that SOC constitutes an important variable that governs the forest $CH_4$ sink. We find that a $CH_4$ sink is enhanced with increasing SOC content on regional and global scales. The revised model with SOC function better reproduces the field observation and estimates a 39% larger global forest $CH_4$ sink (24.27 Tg $CH_4$ $yr^{-1}$) than the model without considering SOC effects (17.46 Tg $CH_4$ $yr^{-1}$). This study highlights the role of SOC in the forest $CH_4$ sink, which shall be factored into future global $CH_4$ budget quantification.

Upland soils are an important sink of atmospheric methane ($CH_4$) owing to the activity of methanotrophs, contributing 5–10% of atmospheric $CH_4$ removal[1,2]. Among upland ecosystems, forests play a pivotal role, accounting for up to 62% of the global soil $CH_4$ sink[3]. Therefore, accurate quantification of the global $CH_4$ sink provided by forest ecosystems is essential to improving our understanding of the global $CH_4$ budget.

The magnitude of global forest $CH_4$ sink remains poorly constrained due to our limited understanding of the process[1,4]. Previous models estimated the global forest $CH_4$ sink ranges from 9 to 24 Tg $CH_4$ $yr^{-1}$[1,3–8], which highlights the necessity for reducing this uncertainty to better estimate the sink. Conventionally, methanotrophs are considered to utilize $CH_4$ as their sole carbon and energy source. Accordingly, the existing estimations using various process-based modeling and meta-analysis approaches generally use soil factors related to gas diffusivity ($CH_4$ and $O_2$ availability) as major determinants of soil $CH_4$ sink[4,6–8]. Along with soil gas diffusivity, other soil variables such as alternative carbon sources[9,10] and nutrient availability[11,12] have been suggested to play a pivotal role in

determining the soil $CH_4$ sink. Incorporation of physiological characteristics of methanotrophs into the process-based model indeed constrained the uncertainty of estimation[7,13,14], suggesting that it is crucial to incorporate them into the model to improve our estimation capability. However, due to our incomplete understanding of the regulatory mechanisms behind these variables, current estimations do not adequately consider these variables in the model[7,13,14]. Thus, investigation of the regulatory mechanisms of the previously overlooked drivers and further consideration of these factors is needed to reduce the uncertainty of global forest $CH_4$ sink estimation.

Soil organic matter (SOM) is a complex of decaying organic matter, degradation products such as humus, and soil microbes, and plays a pivotal role in terrestrial ecosystem functioning[15–17]. While SOM data has traditionally been used as a soil quality indicator in soil science, various disciplines are now acknowledging the significance of SOM attributes as indicators of ecosystem processes, including climate feedback. Therefore, diverse ecosystem process models now incorporate quantitative and qualitative data on SOM[18]. Previous models generally considered SOM to negatively affect the soil $CH_4$ sink

[1]School of Civil and Environmental Engineering, Yonsei University, Seoul 03722, Korea. [2]Global Monitoring Laboratory, National Oceanic and Atmospheric Administration, Boulder, CO, USA. [3]Cooperative Institute for Research in Environmental Sciences, University of Colorado, Boulder, CO, USA. [4]Forest Technology and Management Research Center, National Institute of Forest Science, Gyeonggi, Korea. [5]Warm Temperate and Subtropical Forest Research Center, National Institute of Forest Science, Jeju, Korea. [6]Division of Life Sciences, Korea Polar Research Institute, Incheon 21990, Korea. [7]Department of Earth, Atmospheric, and Planetary Sciences, Purdue University, West Lafayette, IN, USA. ✉e-mail: hj_kang@yonsei.ac.kr

because it can provide carbon substrates to methanogens, thereby increasing $CH_4$ emissions[19]. However, this assumption does not consider the diverse mechanisms of SOM-dependent regulation of soil variables that are associated with soil $CH_4$ oxidation, constraining our estimation capability.

To elucidate the mechanisms underlying the SOM-dependent control of the forest $CH_4$ sink and to further improve global forest $CH_4$ sink estimation, three independent approaches—continuous in-situ measurement, global meta-analysis, and process-based modeling—were used in this study. First, we monitored in-situ $CH_4$ uptake rate and edaphic characteristics, including SOM content, for 2 years in subtropical forests with different tree species (Japanese cedar, oak, mixed forest) and in temperate coniferous forests with different thinning intensities (light, moderate, heavy intensity, and control). Second, we performed a global meta-analysis to establish a relationship between the forest $CH_4$ sink and soil organic carbon (SOC) content on a global scale. Finally, we revised a process-based forest $CH_4$ sink model by considering the effect of SOC content to quantify the global forest $CH_4$ sink.

## Results and discussion
### Field observations

Continuous measurements of $CH_4$ fluxes and edaphic characteristics for two years from April 2018 to March 2020 revealed there are significant positive correlations between SOM content and $CH_4$ uptake rate both in a subtropical forest with different tree species (Fig. 1a; $R^2 = 0.538$, $P < 0.001$) and in a temperate coniferous forest with different thinning intensities (Fig. 1b; $R^2 = 0.258$, $P < 0.001$). The positive relationship observed in temperate forest further suggests that the variability of $CH_4$ uptake between the forests with the same dominant species but different soil characteristics can be partially explained by the SOM content. It is conceivable that the positive correlations observed between SOM content and $CH_4$ uptake rate at the two examined sites may have been primarily influenced by other environmental variables that exhibit variability according to tree species or thinning intensity and that are closely associated with $CH_4$ uptake. However, partial correlation analysis indicates that even after removing the influence of other variables such as air-filled porosity (AFP),

dissolved organic carbon (DOC) content, and water-filled pore space, SOM content remains positively correlated with $CH_4$ uptake rate (Supplementary Table 1). Previous studies have demonstrated that magnitudes of soil $CH_4$ sinks exhibit variability among different tree species, which can be attributed to distinctions in the physical and chemical properties of litter[20–22], and inorganic N content[23]. In addition to the existing literature, our study reveals a noteworthy positive correlation between SOM content and $CH_4$ uptake rate in subtropical forests. This finding further supports the notion that SOM content is an influential factor in distinguishing soil $CH_4$ sink capacity among different tree species. Moderate intensity plot exhibited the highest $CH_4$ uptake rate in temperate forest, which can be attributed to the highest SOM content (Supplementary Fig. 1b) and understory biomass (Supplementary Fig. 2a). We found a strong positive relationship between SOM content and understory biomass (Supplementary Fig. 2b), indicating that understory attributed to higher SOM content[24,25]. These findings suggest that understory vegetation may indirectly promote soil $CH_4$ uptake by increasing SOM content. In addition, understory vegetation can enhance soil $CH_4$ uptake by stimulating oxygen and $CH_4$ transportation to the surface soil layer through the root[26].

On the basis of these field measurement data and statistical analysis, we hypothesize that the following mechanisms underpin the positive effect of SOM on forest $CH_4$ uptake. First, SOM can provide alternative carbon substrates to facultative methanotrophs and enhance their activity by stimulating growth. Soil labile carbon forms such as formate, acetate, and methanol are suggested to increase $CH_4$ oxidation activity by stimulating methanotrophic growth[9,10]. Sullivan et al.[27] performed an in vitro experiment in semiarid soils and reported an increase in $CH_4$ oxidation rate with increasing DOC content. In accordance with previous studies, $CH_4$ uptake rate had significant positive correlations with DOC content both in the subtropical forest and temperate forest at our study sites (Table 1). We also found significant positive correlations between SOM content and DOC content both in the subtropical forest and temperate forest (Table 1). These results indicate that SOM may indirectly stimulate soil $CH_4$ uptake in both types of forests by providing alternative C compounds for methanotrophs.

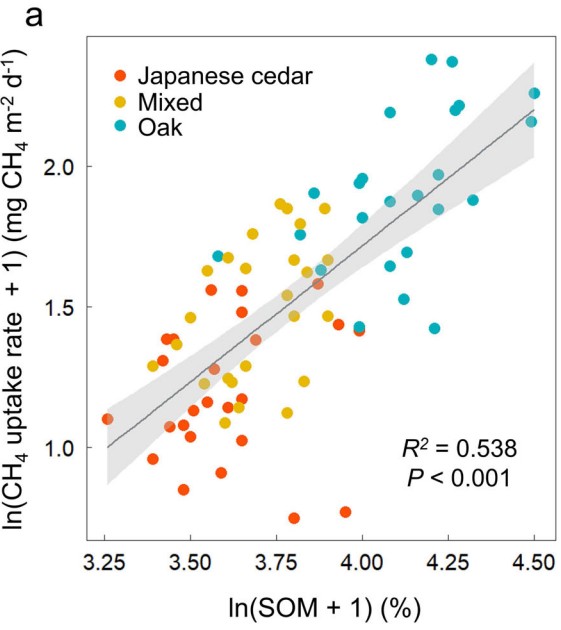
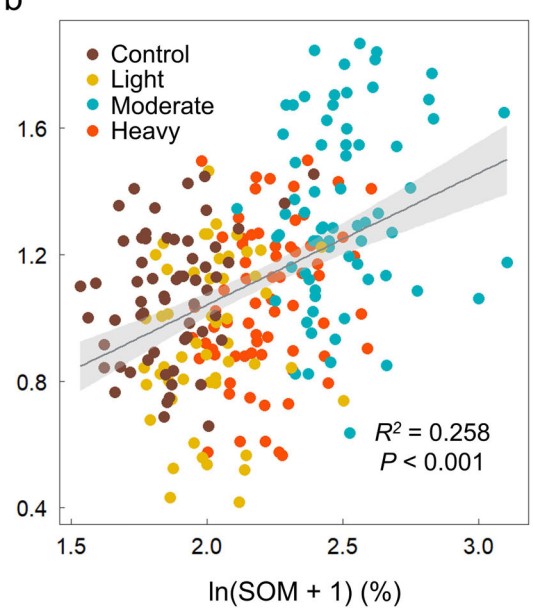

**Fig. 1 | Positive correlations between soil $CH_4$ uptake rate and soil organic matter content.** Relationships between forest soil $CH_4$ uptake rate and soil organic matter (SOM) content in (**a**) subtropical forest with different tree species ($N = 72$, $P < 0.001$) and (**b**) temperate coniferous forest with different thinning intensities ($N = 258$, $P < 0.001$). A linear mixed effect model was used and the error bands surrounding the regression lines represent the 95% confidence interval of the correlation.

**Table 1 | Correlations between CH$_4$ uptake rate and edaphic characteristics**

| | SOM content | | CH$_4$ uptake | |
|---|---|---|---|---|
| | Subtropical forest | Temperate forest | Subtropical forest | Temperate forest |
| SWC | **0.521** | **0.74** | 0.101 (**0.302**) | 0.045 (**0.141**) |
| DOC | **0.802** | **0.367** | **0.631** | **0.231** |
| AFP | 0.04 | **0.268** | **0.497** | **0.547** |

Correlations between SOM content, CH$_4$ uptake rate and edaphic characteristics in subtropical forest ($N = 74$) with different tree species and temperate forest with different thinning intensities ($N = 258$). Values indicate Pearson correlation coefficient and the values in brackets indicate Pearson coefficient when the highly saturated periods are excluded (April 2018 in subtropical forest and December 2018, February 2019 in temperate forest. Seasonal variation of soil water content is presented in Supplementary Fig. 4). Bold numbers indicate statistically significant ($P < 0.05$).
SOM soil organic matter, SWC soil water content, DOC dissolved organic carbon, AFP air-filled porosity.

The second hypothetical mechanism is based on the enhanced water-holding capacity of organic-rich soil[28], which can alleviate water stress and stimulate CH$_4$ oxidation. Generally, soil water content (SWC) has a unimodal relationship with soil CH$_4$ uptake rate[29,30]. Under low soil moisture, water stress can promote ethylene production by plants, which inhibits CH$_4$ oxidation in soil[30,31]. In addition, water stress on methanotrophs can result in reducing their activity[29]. On the other hand, high soil moisture inhibits methanotrophic activity by decreasing gas diffusivity[30]. A global meta-analysis showed that the forest CH$_4$ sink increased with increasing mean annual precipitation, suggesting a higher CH$_4$ uptake at sites with more precipitation on a global scale[32]. These authors suggested that the positive effect of water content induced by water-stress alleviation is larger than the negative effect of decreased CH$_4$ and O$_2$ availability in the forests. Several other studies indicated the prevalent role of precipitation in promoting biological oxidation of CH$_4$ in forest soils[19,33,34], suggesting that enhanced water-holding capacity may stimulate methanotrophic activity, with the exception of highly saturated periods. Although the correlation between SWC and CH$_4$ uptake rate was not significant at our study sites, the highest CH$_4$ uptake rate was observed in the oak forest (Supplementary Fig. 3a) and a plot with moderate intensity thinning (Supplementary Fig. 3b), where the highest SWC was observed in the subtropical forest (Supplementary Fig. 4a) and temperate forest (Supplementary Fig. 4b), respectively. Furthermore, we found significant positive correlations between SWC and CH$_4$ uptake rate when the highly saturated periods were excluded, both in the subtropical forest (Supplementary Fig. 5c; $R^2 = 0.09$; $P < 0.05$), and temperate forest (Supplementary Fig. 5d; $R^2 = 0.02$; $P < 0.05$), indicating that soil moisture positively influences soil CH$_4$ uptake, with the exception of periods of high soil water saturation. Significant positive correlations between SOM content and SWC both in the subtropical forest and temperate forest (Table 1) indicate that higher moisture content in organic matter–rich soil can be attributed to enhanced water-holding capacity. Collectively, these results reinforce our hypothesis that the enhanced water-holding capacity of organic-rich soil may stimulate CH$_4$ oxidation activity in forest soils by alleviating water stress.

Lastly, SOM content increases soil gas transport capability by increasing pore space owing to the positive effect of organic matter on soil aggregation, tilth, and biopore development[15]. Indeed, we found a positive correlation between SOM content and AFP in the temperate forest (Table 1). Although the correlation was not significant in the subtropical forest, the highest AFP was observed in the oak forest, where the SOM content was highest, and the lowest AFP was observed in the Japanese cedar forest, where the SOM content was lowest (Supplementary Fig. 1a and Supplementary Fig. 6a). These results indicate that SOM enhances gas diffusivity in forest soils. AFP was positively correlated with soil CH$_4$ uptake rate both in the subtropical

forest and temperate forest (Table 1), suggesting that SOM may indirectly regulate CH$_4$ uptake by controlling the soil pore space.

In addition to these three suggested mechanisms based on in-situ observations, we speculate that several other mechanisms may play a role in the relationship between SOM and CH$_4$ oxidation rates. For example, increased microbial diversity in SOM-rich soil can increase methanotrophic activity. Recent microbial pure-culture studies revealed that the diversity of heterotrophic bacteria significantly increases the activity and growth rate of methanotrophs by generating products beneficial for methanotrophs or by removing inhibitory products[35–39]. For example, a culture study found the higher CH$_4$ oxidation rate for cultured methanotrophs with diverse heterotrophs compared to samples incubated with methanotrophs only[35]. It was suggested that this mutual relationship was based on the fact that certain heterotrophs produce essential metabolites for methanotrophs, thus stimulating CH$_4$ oxidation activity. As SOM generally increases soil microbial diversity[40,41], a more diverse microbial community in SOM-rich soil can increase CH$_4$ sink capacity by enhancing the positive interaction between methanotrophs and heterotrophic bacteria. Additionally, SOM can provide nutrients such as N and P to methanotrophs and alleviate nutrient limitations[11,12,42–45].

### Relationship between soil CH$_4$ uptake rate and SOC content in global forests

To provide the global context for our in-situ measurements, we retrieved data on mean forest CH$_4$ uptake rate and SOC content worldwide from 81 published articles (Fig. 2a). A significant positive correlation was found between forest mean CH$_4$ uptake rate and SOC content (Fig. 2b; $R^2 = 0.323$, $P < 0.001$), suggesting that SOC indeed positively influences forest CH$_4$ uptake in global forests. In addition, by comparing four ranges of SOC content, we confirmed that CH$_4$ uptake rate increases with the increasing SOC content (Fig. 2c): mean soil CH$_4$ uptake rate was $0.79 \pm 0.09$ mg CH$_4$ m$^{-2}$ d$^{-1}$ in a SOC content range from 0 to 20 g kg$^{-1}$, but it was more than 3 times higher ($2.98 \pm 0.32$ mg CH$_4$ m$^{-2}$ d$^{-1}$) when SOC content ranged from 60 to 100 g kg$^{-1}$. While SOC has been regarded to rather reduce the forest CH$_4$ sink by affecting methanogenesis[19,32], the relationship we found in the field observations and meta-analysis suggest that the positive effect of SOC on soil CH$_4$ uptake outweighs the negative effect at both regional and global scales. These results further suggest the need to incorporate the positive effect of SOC into the estimation model.

### Revised global forest CH$_4$ sink estimation model

We incorporated the effect of SOC into the terrestrial ecosystem model (TEM) to quantify the global forest CH$_4$ sink. Overall, a comparison between the observed and simulated forest CH$_4$ uptake rates showed a 30% reduction of root mean square error (RMSE) values in the new model that included the function of SOC (TEM-SOC; Supplementary Fig. 7b; RMSE = 0.56) in comparison with the model without this function (TEM-DEF; Supplementary Fig. 7a; RMSE = 0.79; Liu et al.[8]). However, when different biomes were analyzed separately, estimation using the new model substantially reduced RMSE in temperate and boreal forests but increased it in the tropical forest. This result indicates that the simulated CH$_4$ uptake rate better represents the observational data in temperate and boreal forests by considering the effect of SOC, but not in the tropical forest.

Overall, TEM-SOC estimates the global forest CH$_4$ sink as $24.27 \pm 1.64$ Tg CH$_4$ yr$^{-1}$, which is 39% larger than the estimate by TEM-DEF ($17.46 \pm 5.09$ Tg CH$_4$ yr$^{-1}$) (Fig. 3). As previous bottom-up estimation using a process-based model underestimated the global upland CH$_4$ sink by 8 Tg CH$_4$ yr$^{-1}$ in comparison with the top-down estimation that uses the atmospheric inverse-modeling framework[46], our revised model reconciles the disparity between the two different approaches. Compared to TEM-DEF, TEM-SOC estimated 38% and 341% higher CH$_4$ sink in temperate forest

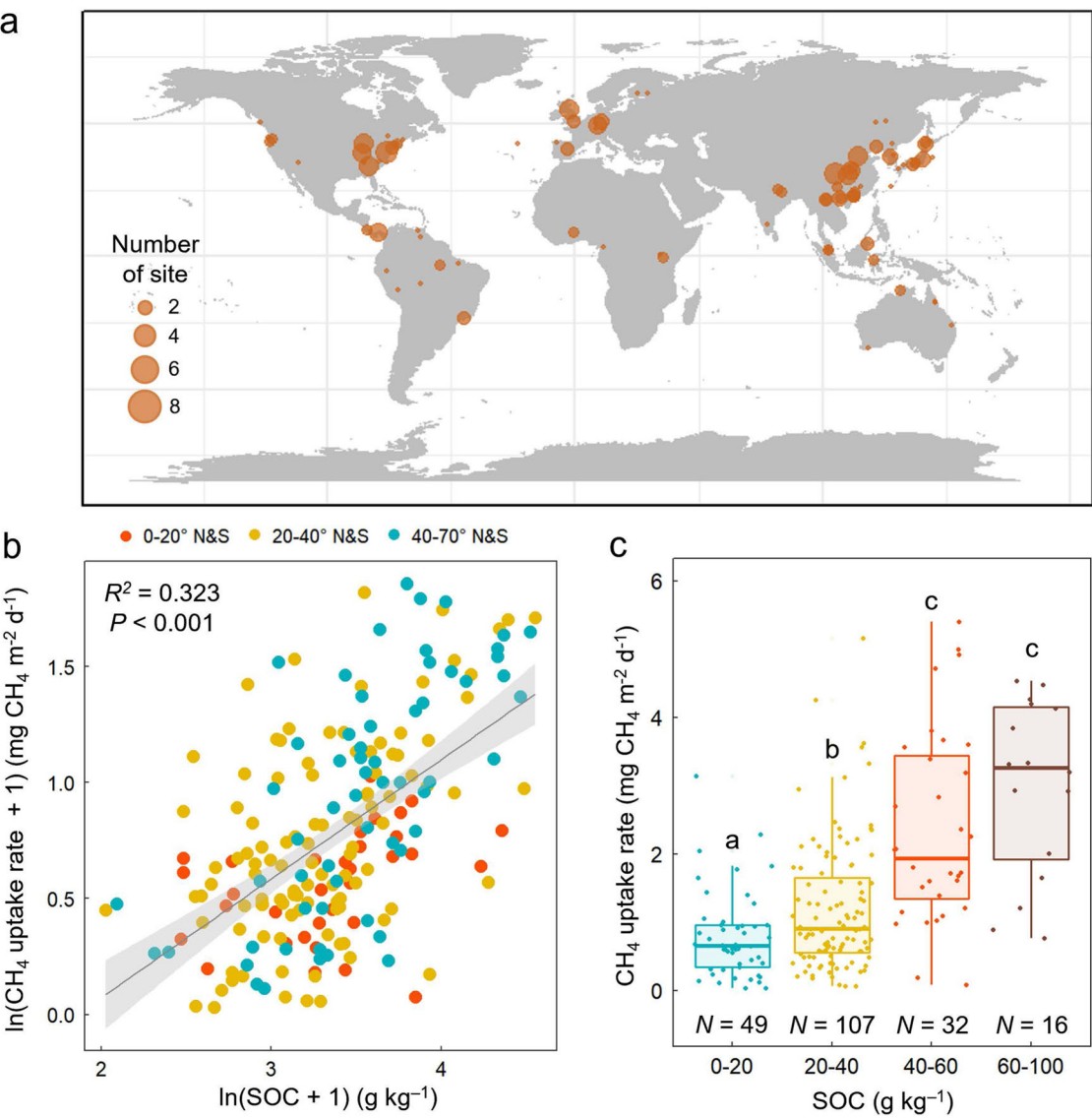

**Fig. 2 | Relationship between soil CH₄ uptake rate and soil organic carbon content based on the selected sites around the globe. a** The location of the 204 sites across the globe that are included in the meta-analysis. The size of the dot represents the number of sites. **b** Relationship between reported forest CH₄ uptake rate and soil organic carbon (SOC) content ($N = 204$, $P < 0.001$). Linear regression was used and the error bands surrounding the regression lines represent the 95%

confidence interval of the correlation. **c** Forest CH₄ uptake rate at different SOC content ranges. Differing letters denote statistically significant differences between SOC content ranges based on the Kruskal–Wallis one-way ANOVA ($P < 0.05$). The thick central line represents the median value, the boxed areas represent the interquartile range, and the whiskers show the maximum and minimum values.

($13.92 \pm 1.44$ Tg CH₄ yr⁻¹ versus $10.08 \pm 3.28$ Tg CH₄ yr⁻¹) and boreal forest ($5.82 \pm 1.27$ Tg CH₄ yr⁻¹ versus $1.32 \pm 0.44$ Tg CH₄ yr⁻¹), respectively, whereas estimated 23% lower CH₄ sink in tropical forest ($4.65 \pm 0.37$ Tg CH₄ yr⁻¹ versus $6.06 \pm 2.8$ Tg CH₄ yr⁻¹) (Fig. 3c). The difference among biomes in the magnitude of responses to SOC incorporation was caused perhaps by different sensitivity of each biome to soil variables including SOC content[32]. In the regions characterized by predominantly low mean temperatures, such as boreal forests, it has been proposed that CH₄ and O₂ availability are not the primary limiting factors for CH₄ uptake[32,33]. Rather, the biological activity of methanotrophs, which may be subject to physiological control by other environmental factors such as nutrient availability and pH, limits the process. Thus, in regions with low mean temperature, the role of soil variables that can promote the biological oxidation of CH₄ physiologically becomes more critical than CH₄ and O₂ availability. The CH₄ sink in temperate and boreal forests estimated by TEM-SOC was larger than that estimated by TEM-DEF because the

biological activity of methanotrophs is limited in these biomes, and TEM-SOC considers the positive physiological effect of SOC on soil CH₄ consumption. Meanwhile, the estimated CH₄ sink in the tropical forest was lower in TEM-DEF than in TEM-SOC because the biological activity of methanotrophs is not limited owing to the high temperature, while the CH₄ and O₂ availability is more likely the limiting factor. Indeed, our global meta-analysis showed that the slope of the linear regression line between SOC content and soil CH₄ uptake rate was steepest at high latitudes (40–70° N&S), followed by middle latitudes (20–40° N&S), and shallowest at low latitudes (0–20° N&S) (Supplementary Fig. 8), indicating that the positive effect of SOC on forest CH₄ sink is weaker in warmer regions. The differences in the magnitudes of responses between biomes may also be induced by different SOC contents. Generally, boreal forests have higher SOC contents than tropical and temperate forests owing to the slow decomposition because of the cold temperature. Because of the high SOC content, the positive effect of SOC in boreal forests results in a substantial increase

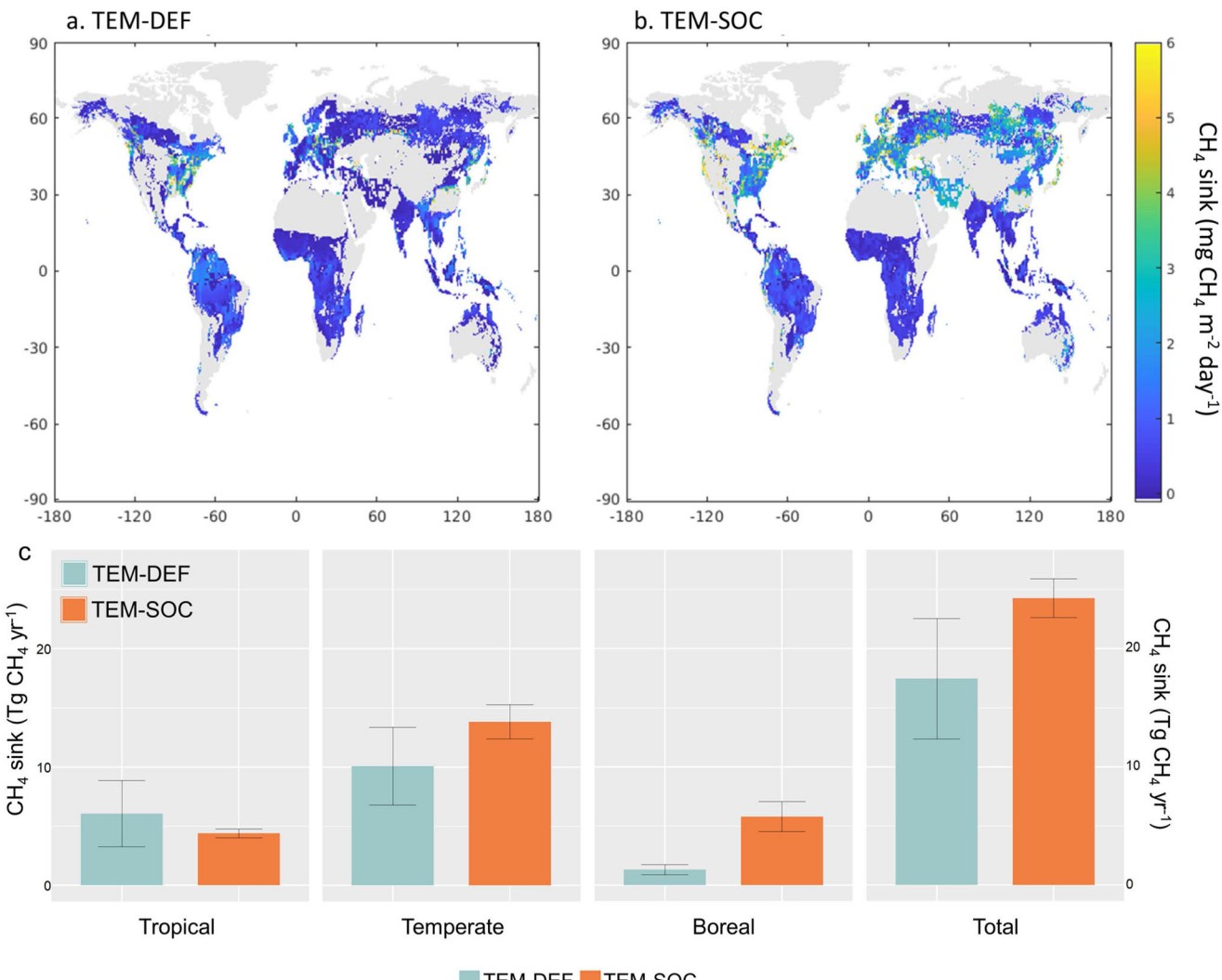

**Fig. 3 | Global forest CH₄ sink estimation.** The simulated global forest mean CH₄ uptake rate of (**a**) the model without the function of SOC (TEM-DEF) and (**b**) the model with the function of SOC (TEM-SOC). **c** Estimated mean annual forest CH₄ sink (Tg CH₄ yr⁻¹) results from TEM-DEF and TEM-SOC during 2000–2016 with different biome types. The error bars represent 1σ of TEM-DEF and TEM-SOC, which were determined by varying the optimized parameters from ensemble simulations (*N* = 50 for TEM-DEF and *N* = 20 for TEM-SOC).

in soil CH₄ sink, while this effect is less significant in tropical forests due to the relatively low SOC content. Process-based soil CH₄ sink model estimates lower CH₄ uptake from the boreal forest than the temperate and tropical forests most likely due to the low temperature in the boreal biome[3,7]. However, previous in-situ observation data consistently suggested that the mean soil CH₄ uptake rate of boreal forests is similar to those of most other biomes[32,34], indicating that previous process-based models underestimated the CH₄ sink capacity of boreal forests.

Our study based on field measurements and global meta-analysis provides evidence on the pivotal role of SOC in the forest soil CH₄ sink on both regional and global scales. Our revised model with SOC function improves the estimation and reconciles the disparity between bottom-up estimation and top-down modeling. Revealing the response of climate feedback to changing SOC dynamics is one of the major challenges for projecting future climate change. SOC stocks in terrestrial ecosystems including forests are sensitive to climate change and are projected to be reduced by global warming[47–51], which may induce a positive feedback to climate change. The global soil CH₄ sink is projected to increase in the future, most likely due to the increase in atmospheric CH₄ concentration and temperature rise[52]. However, our

study suggests that the reduction of the SOC stock in forest ecosystems by warming can partially offset the increasing soil CH₄ sink, potentially inducing positive feedback to warming along with the increasing CO₂ emission. Thus, it is important to incorporate the effect of SOC on the forest CH₄ sink to better understand the impact of climate change and constrain the global CH₄ budget.

## Methods
### Site description and soil sampling
The study sites were located in the Hannam experimental forest (subtropical forest; 33°33′N, 126°65′E) and in Gwangneung experimental forest (temperate forest; 37°76′ E, 127°17′E) in the Republic of Korea. The mean annual temperature was 17.0 °C and 11.7 °C in subtropical forest and temperate forest, respectively, and the mean annual precipitation was 2188 mm and 1364 mm during the observation period in subtropical forest and temperate forest, respectively. In subtropical forest, we selected 3 different tree species: *Quercus acuta* (oak), *Cryptomeria japonica* (Japanese cedar), and mixed forest. In temperate forest, we selected *Pinus koraiensis* dominant forest with 3 different thinning intensities: light intensity (700 trees ha⁻¹), moderate intensity (600 trees ha⁻¹), heavy intensity (500 trees ha⁻¹), and control

(800 trees ha$^{-1}$). Soil samples were collected seasonally (spring, summer, autumn, and winter) in 2018 and 2019 at subtropical forest and monthly from April 2018 to March 2020 at temperate forest. Soil samples were collected to the depth of 0–10 cm after litter layer and O-horizon removal. Soil temperature was measured at the depth of 5 cm after soil sampling. Collected samples were placed in plastic zipper bags and transported to the lab on ice. Samples were stored at 4 °C and further analyses were conducted within 3 days.

## Edaphic characteristics

SWC was measured by oven-drying at 105 °C for 24 h, and SOM content was determined by a loss-on-ignition method. Soil DOC content was extracted by mixing soil with distilled water, filtered through 0.45-μm filter, and then analyzed by a TOC analyzer (TOC-V$_{CHP}$, Shimadzu, Kyoto, Japan). Soil core of 0–10 cm depth was collected to measure bulk density. Based on the measured bulk density, Air-filled porosity was calculated as below:

$$Total\ porosity = (1 - bulk\ density/\rho_s)/100 \qquad (1)$$

$$Air - filled\ porosity = total\ porosity - \theta_v \qquad (2)$$

Where $\rho_s$ is particle density (2.65 g cm$^{-3}$) and $\theta_v$ is volumetric moisture content.

Highly saturated periods were defined as periods when the measurements were taken right after a rain event or snowmelt, and when the water-filled pore space was higher than 85% and 60% in subtropical forests and temperate forests, respectively. Briefly, April 2018 in the subtropical forest and December 2018 and February 2019 in the temperate forest were defined as highly saturated periods.

## Soil CH$_4$ fluxes measurement

Soil CH$_4$ fluxes were measured every 3 months in the subtropical forest from April 2018 to December 2019 and monthly in the temperate forest from April 2018 to March 2020. Fluxes were measured using the closed static chamber method with triplicate measurement in each treatment. Changes in headspace CH$_4$ concentration were measured using GasScouter™ G4301 Mobile Gas Concentration Analyzer (Picarro Inc., Santa Clara, CA, USA) from April 2018 to September 2019. Due to mechanical issues with the portable analyzer, we took the headspace gas sample every 10 min for 50 min and transferred it to a pre-evacuated glass vial from October 2019 to April 2020. We used a GC-FID (CP-3800 Varian, USA) to measure the CH$_4$ concentration. CH$_4$ fluxes were calculated from linear regression slopes (chamber headspace [CH$_4$] vs. time) with a minimum $R^2 = 0.82$.

## Literature review and data collection

We collected 204 data points from 81 published papers using Web of Science and Elsevier's Scopus academic database. We searched articles published between 1989 to 2021 using the keyword '(CH$_4$ uptake OR CH$_4$ sink) AND (forest)'. Reported variables including annual CH$_4$ uptake rate, topsoil SOC or SOM content, mean annual precipitation, mean annual temperature, latitude and longitude were extracted from each paper. Collected SOM contents ($N = 5$) were converted to SOC content by using the simple equation: SOM = SOC × 1.72 (van Bemmelen factor[53]). The van Bemmelen factor has been widely used to convert SOM content to SOC content[54]. We also collected papers that present only total C content ($N = 26$) since inorganic C content in forest soil is negligible[55,56]. When the paper presented SOC values with depth profiles, values below the humus layer were collected. In the case of field manipulation experiments such as warming, elevated CO$_2$, and fertilizer addition, the CH$_4$ uptake rate of only the control plot was collected to prevent other controlling variables affecting the CH$_4$ uptake rate of forests with natural conditions.

## Process-based modeling of global forest CH$_4$ soil sink

TEM is a process-based biogeochemistry model and its CH$_4$, soil, thermal, and hydrological dynamics have been evaluated in previous studies[7,14,57]. The model simulates CH$_4$ production, oxidation, and three transport processes including diffusion, ebullition, and plant-mediated transport between soils and the atmosphere. The version of TEM we used is from Liu et al.[8], which is referred as to TEM-DEF, and we focused on CH$_4$ oxidation process in non-inundated forest regions.

For TEM-DEF, the CH$_4$ oxidation is calculated by the product of the maximum potential oxidation rate ($O_{MAX}$) and limiting functions of methane concentration, soil temperature, soil moisture, redox potential, nitrogen deposition, and diffusion limited by high soil moisture ($C_M$, $T_{SOIL}$, $E_{SM}$, $R_{OX}$, $N_{DP}$, and $D_{MS}$, respectively) (Eq. 3). For the new model, TEM-SOC, we revised the CH$_4$ oxidation function by incorporating the SOC effect on CH$_4$ oxidation ($f(SOC)$ in Eq. 4). We optimized Michaelis-Menten function between SOC and CH$_4$ oxidation using our collected data (Supplementary Fig. 9).

$$M_{O,TEM-DEF}(z,t) = O_{MAX} f(C_M(z,t)) f(T_{SOIL}(z,t)) f(E_{SM}(z,t)) F(R_{OX}(z,t)) f(N_{dp}(z,t)) f(D_{ms}(z,t))$$
$$(3)$$

$$M_{O,TEM-SOC}(z,t) = O_{MAX} f(C_M(z,t)) f(T_{SOIL}(z,t)) f(E_{SM}(z,t)) F(R_{OX}(z,t)) f(N_{dp}(z,t)) f(D_{ms}(z,t)) f(SOC)$$
$$(4)$$

We optimized five parameters related to maximum oxidation potential, soil temperature, and moisture sensitivity for CH$_4$ oxidation related to both TEM-DEF and TEM-SOC for tropical, temperate, and boreal forest regions (Supplementary Table 2 and 3). All other parameters were set the same as in Liu et al.[8]. We used the Shuffled Complex Evolution Approach in R language (SCE-UA-R) to minimize the difference between simulated and observed CH$_4$ oxidation rates[58].

To make spatially- and temporally-varying estimates of CH$_4$ emission and consumption, we used spatially explicit data of land cover, soil pH and textures, meteorology, and leaf area index. We used SOC data from Soilgrids database[59]. The model was applied at the spatial resolution of 0.5° latitude by 0.5° longitude for forest upland ecosystems with a daily time-step during 2000–2016. A year of spin-up was used for CH$_4$ equilibrium in soils, and simulated ecosystem-specific CH$_4$ oxidations were then area weighted for each grid cell. For each site, 20 ensembles were run using SCE-UA-R with 10,000 maximum loops per parameter ensemble, and all of them reached steady state before the end of the loops.

## Statistical analysis

Linear mixed-effect models were used to assess the positive relationship between SOM content and soil CH$_4$ uptake rate in the field. In the model, tree species (subtropical forest) or thinning intensity (temperate forest) was used as a random effect for the incorporation of between-species or between-thinning intensity dissimilarity. Before conducting regression analysis, SOM content and soil CH$_4$ uptake rate was log-transformed to improve the normality of the data and validity of regression analysis. Correlations between soil characteristics (SWC, soil DOC content, and AFP) and SOM content as well as soil characteristics and CH$_4$ uptake rate were assessed with Pearson's product moment correlation coefficients. Linear mixed effect model was used to assess the effect of tree species or thinning intensity on forest soil CH$_4$ uptake rate, SOM content, soil AFPS, and SWC. Tree species or thinning intensity was used as categorical variables, and plot and month were used as random effects. Tukey's HSD tests were used to compare individual means. Partial correlation analysis was conducted to assess whether SOM content was positively correlated with soil CH$_4$ uptake rate without the influences of other variables that vary by tree species or thinning intensity. In this analysis, environmental variables

that (1) vary by tree species or thinning intensity and (2) are closely related to soil $CH_4$ uptake rate were selected as control variables.

Using the data we collected from 81 previously published papers, linear regression analysis was conducted to test whether positive relationship between SOC content and soil $CH_4$ uptake rate also exists in the global forests. Before conducting linear regression analysis, collected soil $CH_4$ uptake rate and SOC content were log-transformed to improve the normality of the data and the validity of the analysis. To assess whether forest soil $CH_4$ uptake increases along the SOC content range, we conducted Kruskal–Wallis one-way ANOVA.

### Reporting summary

Further information on research design is available in the Nature Portfolio Reporting Summary linked to this article.

## Data availability

Supporting Information is available online. The dataset used for meta-analysis can be found in Supplementary Data. The model results are available at the Purdue University Research Repository (PURR) under accession code https://doi.org/10.4231/8K7W-NF84[60].

## Code availability

The code is also archived and freely available at the Purdue University Research Repository (PURR) at https://doi.org/10.4231/8K7W-NF84[60].

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

## Acknowledgements

This study was supported by the Korea Forest Service (2017096A001719BB01), Ministry of Science and ICT of Korea (NRF-2020M1A5A1110494, NRF-2020R1I1A2072824), Ministry of Environment of Korea (2022003640002), and the Ministry of Education of Korea (NRF-2019H1A2A1076239, NRF-2021R1A6A3A03039376, NRF-2019R1A6A 3A01091184, NRF- 2022R1I1A1A01071925).

## Author contributions

J.L. and H.K. conceived and co-designed the study. J.L. conducted field experiments and meta-analysis, and led the writing of the paper. H.K. contributed to writing. Y.O. and Q.Z. built the model. Y.O. conducted the model runs. J.Y., Y.Y., and J.K. contributed to formal analysis and visualization. S.T.L. and Y.O.S. contributed to field survey and writing.

## Competing interests

The authors declare no competing interests.
