## [Peer Review File · Nature Communications]

Soil organic carbon is a key determinant of CH₄ sink in global forest soilsReviewer #1 (Remarks to the Author):

Review of NCOMMS-22-49250-T

General comments

The authors describe a significant relationship between SOC and the CH₄ oxidation capacity (sink) of forest soil. On the basis of a study, a metadata review and modelling work they state SOC has “overarching” control of forest CH₄ oxidation capacity. The authors then suggest the need to integrate SOC data into models estimating forest CH₄ dynamics to produce better understanding of current and future CH₄ budgets at global scales.

This is an important area of research, and I am supportive of study that seeks to extend our capacity to understand the CH₄ sink forest soils represent. The authors have done a reasonable job of exploring the literature, with a few exceptions that will be remarked on below. The text is generally clear and well written, with several ideas well developed and expressed.

Unfortunately, I have several issues with this manuscript as it stands, and do not consider the data sufficient to support the claim that SOC has overarching control of the forest CH₄ sink. Firstly, with regard to the study the authors conducted, the data (as shown in Fig 1) is highly skewed towards the lower end of the x axis, and variation in y appears to increase with greater values of x. I would have expected a detailed description of the statistical techniques used to determine the nature of the data, and the most appropriate model to assess the relationship between SOM and CH₄ uptake; instead the section on statistical analysis is worryingly brief, and does not contain sufficient depth to provide confidence to the reader in the results explored in Fig 1, or the other sections of the MS (metadata, modelling work). Given these results are central to the point of the paper, this needs to be done better. Please note I am not necessarily disputing the findings, it is just that I would need to see more background work done to assure me of their validity to support the concepts the authors are proposing.

With further regard to Fig 1a, I think it is essential for the authors to discuss the potential interaction with the tree species – given the clear effect of species on SOM, I would not be surprised to see that the tree species have affected other aspects of the soil, potentially including the composition of the microbial community. What evidence do we have that this is causation, not just correlation, based on what is presented? I would be very interested in seeing an analysis of the correlation between SOM and CH₄ uptake for each tree species separately – on line 66 it is commented that there is a positive correlation within tree species, but where is the evidence of this? Adding to this, the lack of discussion around the thinning effects needs to be addressed – why has the moderate intensity treatment created such a varied outcome compared to the light or control plots, or even the heavy treatment? Numerically, the extent of the thinning regime is on a shallow gradient, so why the extreme effect? Also, why had the moderate thinning increased SOM more than the heavy thinning? Some discussion is needed.

I also have concerns around how the metadata results are presented in the text. Examination of Fig 2b clearly indicates that the vast majority of the data points are below 40 g C per kg soil, at which point there is no significant relationship with CH₄ uptake (as detailed in Fig 2c). Extending the regression to >75 g C per kg soil is fine as long as the data analysis is done correctly, but I don't see evidence of this as discussed above. Furthermore, splitting the data into categories as in Fig 2c could give the impression of an even distribution of points along the x axis... which is certainly not the case. The data comprising the 40-60 and 60-100 categories comprise perhaps 20% of all the data points, and therefore I question the real-world utility of this figure. I think it could lead the reader to incorrect assumptions, and the disparity in the depth of data for the different categories must be mentioned explicitly.

All things considered, given these issues, I conclude that the authors have not provided sufficient evidence for their claim that SOM exerts “overarching” control of the forest CH₄ sink. I am confident that SOM is a very important factor that does need to be incorporated into models, and so this work is of value for sure, but more work (or adequate explanation of the work that has been done) is required to convince me that it is the dominant factor as stated here.

Specific Comments

Abstract

L4 – long sentence, rephrase for clarity

L9 – considerable, unimpeachable evidence is required to support this claim

Main

L20 – 5-10% - what is this variation driven by? Environmental factors, or instability in the data from measure to measure?

L20 – link these sentences together more clearly. Disjointed currently.

L24 – improve the phrasing – this sentence is not clear.

L26 – why are the uncertainties so large? What is the driver?

L29 – not sure bookkeeping is the right term here

L32 - not sure controlling is the right word in this context. Determining?

L33- 36 – rephrase for clarity.

L37 – definitely not unrecognised drivers – just not part of global models yet.

L24-38 – the point of this paragraph seems to be the need to reduce uncertainty – start with this point, rather than end with it.

L40 – are soil microbes part of the SOM? Or do they regulate it? See

<https://www.nature.com/articles/s43705-021-00071-7>

L41-44 – messy sentence, needs rephrasing

Field Observations

L63 and many other places – if you state the correlation is positive, please just report the r^2 value and not the R. More useful to see the degree of explanation if you state the direction of the correlation, which is consistently done in the text.

L66 – I am not convinced that within tree species the CH₄ oxidation rate was correlated with SOM – what is the P and R² value for each species?

L78-80 – Need data on this to move beyond speculation... stable isotope work to show which C molecules are fuelling methanotroph activity more than others etc

L84 – not fully accurate – Zhou et al. proposed that the inhibitory effect of ethylene released by drought stressed plants was a significant driver, not the effect of drought on methanotrophs.

L88-92 – can you discuss this in relation to the findings in Science of the Total Environment 757 (2021) 144089

L99 – what defines highly saturated? Seems to be 1/6 of the time for temperate forests, so not uncommon.

L105-107 – how? Work with ethylene and drought stress may provide a pathway, but the work on this to date shows it is a 2nd order effect well below the potential for too much soil moisture to reduce gas diffusion into soil.

L122 – pure-culture studies in the context of microbial diversity...? Some additional explanation is needed please.

Relationship between soil CH₄ uptake rate and SOC content in global forests

L136 – I am not a fan of Fig 2c, as described above.

L141 – This sentence need to be rephrased. Also, the content has been mentioned earlier in the text.

Revised global forest CH₄ sink estimation model

Figs 3a and 3b are not cited in the text.

L160 – define bottom up and top down estimation

L162-166 – what were the significance values? Are you talking about changes in the amount of the uncertainty, or changes in the means values for the different biomes? I see changes in uncertainty, but not the mean.

168-170 – this sentence needs to be rewritten for clarity.

Methods

L322 – The thinning intensity is a very shallow gradient, and I think that more extreme gradients would have created more useful outcomes. However, I assume you were using what was available

to you at the time, and could not choose this. However, 300, 500, 800 and 1100 stems per ha would have been far more interesting to assess.

L350 – with, not of

L359 – “forest” as a search term covers many things... planted, natural, exotic, indigenous etc. I think you could gain some further value for your study by looking at different forest types as this would add another context to your results – if the number of replicates allows.

L362 – define the source of 1.72

L364 – sometimes carbonate in forest soils is not negligible – e.g. Applied Spectroscopy

64(10):1167-75 and others

L408-415 – much more detail is required to communicate why the tests you used were chosen, and how you tested their inherent assumptions against the data you have to make sure they were the best choice.

Simeon Smail

Reviewer #2 (Remarks to the Author):

In this manuscript, the authors stressed the dominant role of soil organic carbon for CH₄ sinks in global forest soils, which is very sceptical. Firstly, methane oxidizers get the majority of C from CH₄ being oxidized, not SOC. Secondly, the main results are inconsistent. The SOC-Model resulted in an increase in CH₄ sink for the temperate forests but a decrease in the subtropical forest, which was inconsistent with the observation as DOC showed a much higher positive correlation with CH₄ uptake in the subtropical forest compared with that in the temperate forest. In lines 142-145, which data (observations and metadata) could suggest that the positive effect of SOC on soil CH₄ uptake is larger than the negative effect on both regional and global scales? How? In lines 166-172, previous studies have reported high methane uptake rates with low temperatures (even below zero) and low temperatures limited methane uptake mainly by freezing soil, thus reducing the availability of soil water and O₂. How could high SOC promote CH₄ uptake in this situation? The microbiological data evidence is required to support this point. High heterotrophs activities in SOC-rich soil may benefit methanotroph growth and activity, but it should not be the key determinant.

Reviewer #3 (Remarks to the Author):

This manuscript reports experiments and a global analysis of published data to suggest that the ability of forest soil to be a methane sink is positively correlated with SOC content. They suggest this is largely due to greater soil wetting with higher SOC but may also be due to higher microbial diversity, direct interaction with organic matter such as acetate and methanol, or increase methanogenesis. I find this paper important, well-written, and compelling. My only comment is that the section starting at line 168 has a long discussion of high and low temperatures. It would really help to know what values are considered to be high and low in this context.

Note: Reviewers' comments are in black italic and responses are in blue color.

REVIEWER COMMENTS

Reviewer #1 (Remarks to the Author):

Review of NCOMMS-22-49250-T

General comments

The authors describe a significant relationship between SOC and the CH₄ oxidation capacity (sink) of forest soil. On the basis of a study, a metadata review and modelling work they state SOC has “overarching” control of forest CH₄ oxidation capacity. The authors then suggest the need to integrate SOC data into models estimating forest CH₄ dynamics to produce better understanding of current and future CH₄ budgets at global scales.

This is an important area of research, and I am supportive of study that seeks to extend our capacity to understand the CH₄ sink forest soils represent. The authors have done a reasonable job of exploring the literature, with a few exceptions that will be remarked on below. The text is generally clear and well written, with several ideas well developed and expressed.

Response: We thank you for taking the time to review our manuscript and provide insightful comments. We made a concerted effort to adequately respond to each suggestion as below.

1. Unfortunately, I have several issues with this manuscript as it stands, and do not consider the data sufficient to support the claim that SOC has overarching control of the forest CH₄ sink. Firstly, with regard to the study the authors conducted, the data (as shown in Fig 1) is highly skewed towards the lower end of the x axis, and variation in y appears to increase with greater values of x.

Response: The reviewer made a valid point. It appears that the SOM content data for temperate forests is right-skewed, which is likely due to three outlier points measured in the 'moderate intensity' plot. The skewness and kurtosis for the SOM content in temperate forests are 1.3 and 2.9, respectively (Fig. R1). However, when the outliers are removed, the skewness and kurtosis decrease to 0.69 and 0.03, respectively (Fig. R1). These results suggest that the right-skewed nature of the SOM content data in temperate forests is mainly attributable to those outliers.

Fig. R1. Distribution histogram of SOM content data in temperate forest with all data included (left) and 3 outlier points excluded (right).

Hair et al. (2010) and Bryne (2010) proposed that data exhibiting skewness between -2 to 2 and kurtosis between -7 to 7 are regarded as normally distributed, and meet the statistical assumption of normality. As a general guideline, a skewness between -1 and $+1$ is considered excellent, but a value between -2 and $+2$ is typically deemed

acceptable (Hair et al., 2022). George & Mallery (2010) also suggested that the values for skewness between -2 and $+2$ are considered acceptable in order to prove normal univariate distribution. Based on these assumptions, the distribution of SOM content data in temperate forest can be considered a normal distribution and the result of regression analysis is valid. To improve the normality of data and minimize the impact of skewed data on regression model, data were log-transformed, and the skewness and kurtosis were examined as follows:

Fig. R2. Distribution histogram of log-transformed SOM content data in temperate forest with all data included (left) and 3 outlier points excluded (right).

As presented in Fig. R2, distribution is much improved by log-transformation, with a skewness of 0.34 and kurtosis of -0.14 with all data, and skewness of 0.12 and kurtosis of -0.71 without outliers.

SOM content data of subtropical forest were less right-skewed compared to temperate forest where skewness was 1.06 and kurtosis was 0.77 (Fig. R3). Similar to temperate forest, SOM content data in subtropical forest contain two outlier points measured in

oak forest. Upon excluding these outliers, skewness and kurtosis were improved to 0.73 and -0.43 , respectively (Fig. R3).

Fig. R3. Distribution histogram of SOM content data in subtropical forest with all data included (left) and 3 outlier points excluded (right).

As previously stated, the distribution of SOM data in subtropical forest with a skewness of 1.06 and kurtosis of 0.77 can be regarded as a normal distribution, and the result of regression analysis is valid. To further improve the normality of data and minimize the impact of skewed data on the regression model, data were log-transformed, and the skewness and kurtosis were assessed as follows:

Fig. R4. Distribution histogram of log-transformed SOM content data in subtropical forest with all data included (left) and 3 outlier points excluded (right).

Similar to temperate forests, log-transformation of SOM content significantly improves normality in subtropical forests, with a skewness of 0.49 and kurtosis of -0.49 with all data, and skewness of 0.33 and kurtosis of -0.84 without outliers (Fig. R4).

We have verified that log-transformation enhances the normality of SOM content data in both sites. Likewise, we have also confirmed that log-transformation improves the normality of CH_4 uptake rate as follows:

Subtropical forest

Temperate forest

Fig. R5. Distribution histograms of CH₄ uptake rates, both pre- and post-log transformation, in the subtropical and temperate forests.

Therefore, to improve the normality of the data and validity of the linear regression analysis, Fig. 1 has been modified by log-transforming both SOM content and soil CH₄ uptake rate. Furthermore, we evaluated the relationship between soil CH₄ uptake rate and SOM content by using a linear mixed-effect model, which incorporated between-species dissimilarity for subtropical forests and between-thinning intensity dissimilarity for temperate forests. In the model, tree species or thinning intensity was used as a random effect to minimize the risk of false positives.

Fig. 1 before modification (original version)

Fig. 1 after modification (revised version)

References

Byrne, B. M. *Structural equation modeling with AMOS: Basic concepts, applications, and programming*. New York: Routledge. (2010).

George, D., & Mallery, M. *SPSS for Windows Step by Step: A Simple Guide and Reference, 17.0 update (10a ed.)* Boston: Pearson. (2010).

Hair, J., Black, W. C., Babin, B. J. & Anderson, R. E. *Multivariate data analysis* (7th ed.). Upper Saddle River, New Jersey: Pearson Educational International. (2010).

Hair, J. F., Hult, G. T. M., Ringle, C. M., & Sarstedt, M. *A Primer on Partial Least Squares Structural Equation Modeling (PLS-SEM)* (3 ed.). Thousand Oaks, CA: Sage. (2022).

2. I would have expected a detailed description of the statistical techniques used to determine the nature of the data, and the most appropriate model to assess the relationship between SOM and CH₄ uptake; instead the section on statistical analysis is worryingly brief, and does not contain sufficient depth to provide confidence to the reader in the results explored in Fig 1, or the other sections of the MS (metadata, modelling work). Given these results are central to the point of the paper, this needs to be done better. Please note I am not necessarily disputing the findings, it is just that I would need to see more background work done to assure me of their validity to support the concepts the authors are proposing.

Response: We agree with the reviewer that the explanation of the statistical analysis was rather succinct. Therefore, we have included additional information on the statistical analysis to provide more comprehensive support for our findings, as presented below:

Line 331-352: “Linear mixed-effect models were used to assess the positive relationship between SOM content and soil CH₄ uptake rate in the field. In the model, tree species (subtropical forest) or thinning intensity (temperate forest) was used as a random effect for the incorporation of between-species or between-thinning intensity dissimilarity. Before conducting regression analysis, SOM content and soil CH₄ uptake rate was log-transformed to improve the normality of the data and validity of regression analysis. Correlations between soil characteristics (SWC, soil DOC content, and AFP) and SOM content as well as soil characteristics and CH₄ uptake rate were assessed with Pearson’s product moment correlation coefficients. Linear mixed effect model was used to assess the effect of tree species or thinning intensity on forest soil CH₄ uptake

rate, SOM content, soil AFPS, and SWC. Tree species or thinning intensity was used as categorical variables, and plot and month were used as random effects. Tukey's HSD tests were used to compare individual means. Partial correlation analysis was conducted to assess whether SOM content was positively correlated with soil CH₄ uptake rate without the influences of other variables that vary by tree species or thinning intensity. In this analysis, environmental variables that 1) vary by tree species or thinning intensity and 2) are closely related to soil CH₄ uptake rate were selected as control variables.

Using the data we collected from 81 previously published papers, linear regression analysis was conducted to test whether positive relationship between SOC content and soil CH₄ uptake rate also exists in the global forests. Before conducting linear regression analysis, collected soil CH₄ uptake rate and SOC content were log-transformed to improve the normality of the data and the validity of the analysis. To assess whether forest soil CH₄ uptake increases along the SOC content range, we conducted Kruskal–Wallis one-way ANOVA.”

3. With further regard to Fig 1a, I think it is essential for the authors to discuss the potential interaction with the tree species – given the clear effect of species on SOM, I would not be surprised to see that the tree species have affected other aspects of the soil, potentially including the composition of the microbial community. What evidence do we have that this is causation, not just correlation, based on what is presented?

Response: The reviewer made a valid point. The observed positive correlations between SOM content and CH₄ uptake rate in both subtropical and temperate forests may be attributed to the influence of other environmental variables that vary by tree species or thinning intensity. To provide evidence that the positive relationship between SOM content and CH₄ uptake rate is valid even after the removal of the influences of other variables that are varied by tree species or thinning intensity, we conducted a partial correlation analysis. We selected control variables that exhibit correlation with both SOM content and forest CH₄ uptake rate. Briefly, we found that

the correlations between SOM content and CH₄ uptake rate remain significant after the removal of the influence of the control variables. Regarding the result of the partial analysis, Supplementary Table 1 has been newly added as follows:

Supplementary Table 1. Result of partial correlation analysis between SOM content and soil CH₄ uptake rate in subtropical forest and temperate forest (AFP: air-filled porosity DOC: dissolved organic carbon, WFPS: water-filled pore space).

Control variables	Study site	Correlation coefficient	P -value
None	Subtropical forest	0.757	<0.001
	Temperate forest	0.494	<0.001
AFP, DOC, WFPS	Subtropical forest	0.645	<0.001
	Temperate forest	0.390	<0.001

In response to the reviewer's comment, we have included the following sentences to the manuscript:

Line 67-74: "It is conceivable that the positive correlations observed between SOM content and CH₄ uptake rate at the two examined sites may have been primarily influenced by other environmental variables that exhibit variability according to tree species or thinning intensity and that are closely associated with CH₄ uptake. However, partial correlation analysis indicates that even after removing the influence of other variables such as air-filled porosity (AFP), dissolved organic carbon (DOC) content, and water-filled pore space, SOM content remains positively correlated with CH₄ uptake rate (Supplementary Table 1)."

In addition, we analyzed methanotrophic composition data using 16S rRNA sequencing. We found a significant separation of methanotrophic community structure

based on tree species (Fig. R6). Interestingly, SOM was found to be most closely associated with the methanotrophic community, explaining 34% of the variation in community structure. This finding further reinforces our conclusion that SOM content constitutes a significant variable that distinguishes soil CH₄ uptake rate across tree species within our study site.

Fig. R6. Canonical correspondence analysis (CCA) of the methanotrophic community structure (b). Soil organic matter (SOM) content, soil water content (SWC), dissolved TN, and NO₃⁻ were significantly correlated with methanotrophic community structure ($P < 0.05$). Only statistically significant environmental variables were presented with vector arrow.

Table R1. The result of canonical correspondence analysis. Statistically significant R^2 values are written in bold ($P < 0.05$). (SOM: soil organic matter, DOC: dissolved organic carbon, TN: total nitrogen, SWC: soil water content, TC: total carbon)

Variables	R^2	p -value
SOM	0.34	< 0.001
DOC	0.05	0.016
TN	0.05	0.016
TC	0.04	0.061

NO ₃ ⁻	0.12	0.001
NH ₄ ⁺	0.01	0.246
SWC	0.1	0.004
pH	0.01	0.272

4. I would be very interested in seeing an analysis of the correlation between SOM and CH₄ uptake for each tree species separately – on line 66 it is commented that there is a positive correlation within tree species, but where is the evidence of this?

Response: As the reviewer suggested, the relationship between SOM content and CH₄ uptake rate was analyzed separately for each species as follows:

Fig. R7. Relationship between SOM content and soil CH₄ uptake rate with different tree species in subtropical forest.

We found significant positive correlations between SOM content and CH₄ uptake rate in mixed forest and oak forest whereas the relationship was not significant in cedar forest.

What we intended to explain by “within species” was that within forests with the same dominant tree species but different environmental conditions, including tree density

and soil characteristics, SOM content is one of the controlling variables that could differentiate soil CH₄ uptake rate. This is supported by the positive correlation between SOM content and soil CH₄ uptake rate observed in the temperate forest, where the dominant species was *Pinus Koraiensis* while the tree density and soil characteristics differed significantly between sites. We recognize that sentence L65-66 could be misleading and revised it as follows:

Line 65-67: “The positive relationship observed in temperate forest further suggests that the variability of CH₄ uptake between the forests with the same dominant species but different soil characteristics can be partially explained by the SOM content.”

5. Adding to this, the lack of discussion around the thinning effects needs to be addressed – why has the moderate intensity treatment created such a varied outcome compared to the light or control plots, or even the heavy treatment? Numerically, the extent of the thinning regime is on a shallow gradient, so why the extreme effect? Also, why had the moderate thinning increased SOM more than the heavy thinning? Some discussion is needed.

Response: The highest SOM content in the moderate intensity plot can be attributed to the growth of understory vegetation, which was confirmed by biomass measurement of shrubs and herbaceous plants. We found a strong positive relationship between SOM content and understory biomass (now Supplementary Fig. 2), indicating that understory vegetation can stimulate CH₄ sink capacity by increasing SOM content. Understory is known to increase SOM content by means of augmenting litter input to soil (Yang et al., 2018; Fang et al., 2021). A meta-analysis on the impact of understory removal on SOM content also demonstrated that understory removal significantly decreases soil organic carbon content most likely due to reduced carbon input (Zhang et al., 2022). In addition, Plain et al. (2019) postulated that the presence of understory vegetation may potentially heighten soil CH₄ uptake rates by facilitating the transfer of both oxygen and CH₄ to the uppermost layer of soil through the root system. This suggests that understory vegetation in moderate intensity plot may also contribute to the stimulation of CH₄ uptake by increasing gas transportation to the soil.

We added following sentences to further explain the effect of thinning as follows:

Line 80-87: “Moderate intensity plot exhibited the highest CH₄ uptake rate in temperate forest, which can be attributed to the highest SOM content (Supplementary Fig. 1b) and understory biomass (Supplementary Fig. 2a). We found a strong positive relationship between SOM content and understory biomass (Supplementary Fig. 2b), indicating that understory attributed to higher SOM content^{24,25}. These findings suggest that understory vegetation may indirectly promote soil CH₄ uptake by increasing SOM content. In addition, understory vegetation can enhance soil CH₄ uptake by stimulating oxygen and CH₄ transportation to the surface soil layer through the root²⁶.”

Supplementary Fig. 2. Understory biomass with different thinning intensity (a) and the relationship between understory biomass and SOM content (b) in temperate forest.

In addition, we added a brief discussion on the findings from the subtropical forest as follows:

Line 74-80: “Previous studies have demonstrated that magnitudes of soil CH₄ sinks exhibit variability among different tree species, which can be attributed to distinctions in the physical and chemical properties of litter^{20–22}, and inorganic N content²³. In

addition to the existing literature, our study reveals a noteworthy positive correlation between SOM content and CH₄ uptake rate in subtropical forests. This finding further supports the notion that SOM content is an influential factor in distinguishing soil CH₄ sink capacity among different tree species.”

References

- Fang, X. M. et al. Litter addition and understory removal influenced soil organic carbon quality and mineral nitrogen supply in a subtropical plantation forest. *Plant Soil* **460**, 527–540 (2021).
- Plain, C., Ndiaye, F. K., Bonnaud, P., Ranger, J. & Epron, D. Impact of vegetation on the methane budget of a temperate forest. *New Phytol.* **221**, 1447–1456 (2019).
- Yang, Y. et al. Understory vegetation plays the key role in sustaining soil microbial biomass and extracellular enzyme activities. *Biogeosciences* **15**, 4481–4494 (2018).
- Zhang, S. et al. A meta-analysis of understory plant removal impacts on soil properties in forest ecosystems. *Geoderma* **426**, 116116 (2022).

6. *I also have concerns around how the metadata results are presented in the text. Examination of Fig 2b clearly indicates that the vast majority of the data points are below 40 g C per kg soil, at which point there is no significant relationship with CH₄ uptake (as detailed in Fig 2c). Extending the regression to >75 g C per kg soil is fine as long as the data analysis is done correctly, but I don't see evidence of this as discussed above.*

Response: The reviewer made a valid point. Since forest soils are relatively mineral in their characteristics compared to organic-rich environments such as peatland or marsh, the majority of SOC content observed in previous studies concentrated below 40 g C per kg soil. Indeed, the distribution of the global forest SOC dataset we used for the process-based modeling indicates that the majority of the SOC contents are found to be below 40 g C per kg soil (Fig. R8). Thus, we believe that the dataset

employed to assess the relationship between SOC content and forest CH₄ uptake rate reflects the nature of global forests' SOC distribution.

Fig. R8. Distribution histograms of global forest SOC content (5-15cm) obtained from Soilgrids.

In response to the reviewer's feedback, we conducted regression analyses on two separate ranges of SOC content: 0-40 g C per kg soil and 40-100 g C per kg soil (Fig. R9). The results revealed significant correlations between SOC content and CH₄ uptake rate in both ranges, indicating the validity of positive relationship between SOC content and CH₄ uptake rate regardless of the SOC content range.

Fig. R9. Relationship between CH₄ uptake rate and SOC content in different SOC content ranges.

We then evaluated whether the normality of the SOC content dataset could be improved by applying a log-transformation. The results showed that log-transformation produced a well-normalized distribution of SOC content, as demonstrated below:

Fig. R10. Distribution histograms of forest SOC content before and after log-transformation.

We also found the improved normality of the CH₄ uptake rate after log-transformation as follows:

Fig. R11. Distribution histograms of forest soil CH₄ uptake rate before and after log-transformation.

Therefore, we performed a regression analysis using the log-transformed values of SOC content and CH₄ uptake rate, and the Fig.2b has been revised as follows:

7. Furthermore, splitting the data into categories as in Fig 2c could give the impression of an even distribution of points along the x axis... which is certainly not the case. The data comprising the 40-60 and 60-100 categories comprise perhaps 20% of all the data points, and therefore I question the real-world utility of this figure. I think it could lead the reader to incorrect assumptions, and the disparity in the depth of data for the different categories must be mentioned explicitly.

Response: We agree with the reviewer. Omitting the number of data points for each bar in the plot may lead to misinterpretation by readers. Therefore, we modified Fig. 2c by presenting it as a box plot with individual data points and the number of data points, as illustrated below:

Fig. 2c. CH₄ uptake rate with different SOC content ranges.

8. All things considered, given these issues, I conclude that the authors have not provided sufficient evidence for their claim that SOM exerts “overarching” control of the forest CH₄ sink. I am confident that SOM is a very important factor that does need to be incorporated into models, and so this work is of value for sure, but more work (or adequate explanation of the work that has been done) is required to convince me that it is the dominant factor as stated here.

Response: We acknowledge that the use of the term "overarching" may appear excessive given that there are numerous other environmental variables that are also intimately associated with the forest CH₄ sink. Therefore, we revised the sentence that contains the word "overarching" as follows:

Line 7-9: "Here, using *in-situ* observations, global meta-analysis, and process-based modeling, we provide evidence to support the notion that SOC constitutes an important variable that governs the forest CH₄ sink."

Specific Comments

Abstract

9. L4 – long sentence, rephrase for clarity

Response: The reviewer made a valid point. The sentence has been revised as follows:

Line 4-7: "Soil organic carbon (SOC) is a primary regulator of the forest–climate feedback. However, its indicative capability for the soil CH₄ sink is poorly understood due to the incomplete knowledge of the underlying mechanisms. Therefore, SOC is not explicitly included in the current model estimation of the global forest CH₄ sink."

10. L9 – considerable, unimpeachable evidence is required to support this claim

Response: Please see the response to comment 8.

Main

11. L20 – 5-10% - what is this variation driven by? Environmental factors, or instability in the data from measure to measure?

Response: The variation is mainly from model structure and parameter uncertainty

(Saunois et al., 2020). Saunois et al. (2020) estimated global methane soil sinks of 28 Tg CH₄ yr⁻¹ (9–47 Tg CH₄ yr⁻¹), and the results are reported by Zhuang et al. (2013), Kirschke et al. (2013), and Saunois et al. (2016). Considering structural uncertainty in the various models' assumptions and parameters, Saunois et al. (2020) further revised and reported the median and range of Tian et al. (2016): 30 [11–49] Tg CH₄ yr⁻¹ for the periods 2000–2009 and 2008–2017.

References

Kirschke, S. et al. Three decades of global methane sources and sinks. *Nat. Geosci.* **6**, 813–823 (2013).

Saunois, M. et al. The global methane budget 2000–2012. *Earth Syst. Sci. Data* **8**, 697–751 (2016).

Saunois, M. et al. The Global Methane Budget 2000 – 2017. *Earth Syst. Sci. Data* **12**, 1561–1623 (2020).

Tian, H. et al. The terrestrial biosphere as a net source of greenhouse gases to the atmosphere. *Nature* **531**, 225–228 (2016).

Zhuang, Q. et al. Response of global soil consumption of atmospheric methane to changes in atmospheric climate and nitrogen deposition. *Global Biogeochem. Cycles* **27**, 650–663 (2013).

12. L20 – link these sentences together more clearly. Disjointed currently.

Response: The sentences have been revised as follows:

Line 18–20: “Upland soils are an important sink of atmospheric methane (CH₄) owing to the activity of methanotrophs, contributing 5–10% of atmospheric CH₄ removal^{1,2}. Among upland ecosystems, forests play a pivotal role, accounting for up to 62% of the global soil CH₄ sink³”

13. L24 – improve the phrasing – this sentence is not clear.

Response: The sentence has been revised as follows:

Line 23-24: “The magnitude of global forest CH₄ sink remains poorly constrained due to our limited understanding of the process^{1,4}.”

14. L26 – why are the uncertainties so large? What is the driver?

Response: The estimation and uncertainty are from process-based modeling and meta-analysis. For a process-based modeling approach, uncertainties are from model assumptions, structure, and parameterization. For meta-analysis approach, uncertainties are from data uncertainty and extrapolation (Saunois et al. 2020).

Reference

Saunois, M. *et al.* The Global Methane Budget 2000 – 2017. *Earth Syst. Sci. Data* **12**, 1561–1623 (2020).

15. L29 – not sure bookkeeping is the right term here

Response: The reviewer made a valid point. The sentence has been revised as follows:

Line 27-29: “Accordingly, the existing estimations using various process-based modeling and meta-analysis approaches generally use soil factors related to gas diffusivity (CH₄ and O₂ availability) as major determinants of soil CH₄ sink^{4,6–8}.”

16. L32 - not sure controlling is the right word in this context. Determining?

Response: The word has been revised to ‘determining’ as reviewers’ suggestion.

17. L33- 36 – rephrase for clarity.

Response: The sentences have been revised as follows:

Line 32-38: “Incorporation of physiological characteristics of methanotrophs into the process-based model indeed constrained the uncertainty of estimation^{7,13,14}, suggesting that it is crucial to incorporate them into the model to improve our estimation capability. However, due to our incomplete understanding of the regulatory mechanisms behind these variables, current estimations do not adequately consider these variables in the model^{7,13,14}.”

18. L37 – definitely not unrecognised drivers – just not part of global models yet.

Response: The sentence has been revised as follows:

Line 36-38: “Thus, investigation of the regulatory mechanisms of the previously overlooked drivers and further consideration of these factors is needed to reduce the uncertainty of global forest CH₄ sink estimation.”

19. L24-38 – the point of this paragraph seems to be the need to reduce uncertainty – start with this point, rather than end with it.

Response: We added a sentence to the second sentence of the paragraph as follows:

Line 24-26: “Previous models estimated the global forest CH₄ sink ranges from 9 to 24 Tg CH₄ yr⁻¹ (1,3–8), which highlights the necessity for reducing this uncertainty to better estimate the sink.”

20. L40 – are soil microbes part of the SOM? Or do they regulate it? See <https://www.nature.com/articles/s43705-021-00071-7>

Response: Soil organic matter contains the living part of soil including soil microbes. We, therefore, believe that soil microbes are components of SOM and also have a

regulatory function.

Here is a reference:

“Soil organic carbon is composed of soil microbes, decaying organic matter and degradation products like humus.”

Soil carbon unearthed. *Nat. Geosci.* **13**, 523 (2020). <https://doi.org/10.1038/s41561-020-0624-z>

21. L41-44 – messy sentence, needs rephrasing

Response: The sentence has been revised as follows:

Line 41-44: “While SOM data has traditionally been used as a soil quality indicator in soil science, various disciplines are now acknowledging the significance of SOM attributes as indicators of ecosystem processes, including climate feedback. Therefore, diverse ecosystem process models now incorporate quantitative and qualitative data on SOM¹⁸.”

Field Observations

22. L63 and many other places – if you state the correlation is positive, please just report the r^2 value and not the R . More useful to see the degree of explanation if you state the direction of the correlation, which is consistently done in the text.

Response: The reviewer made a valid point. R values based on regression analysis reported in the manuscript have been revised to R^2 value.

23. L66 – I am not convinced that within tree species the CH₄ oxidation rate was correlated with SOM – what is the P and R^2 value for each species?

Response: Please see the response to comment 4.

24. L78-80 – Need data on this to move beyond speculation... stable isotope work to show which C molecules are fuelling methanotroph activity more than others etc

Response: The form of C substrate that can stimulate methanotrophic activity would be different depending on the methanotrophic community composition of soils. We collected the manuscripts describing the physiological characteristics of facultative methanotrophs and summarize them as follows:

Table R2. List of facultative methanotrophs and their preferential C form.

Species	Stain	Family	Multi carbon utilized	Reference
Methylocystis bryophila	H2s	Methylocystaceae	Acetate	Belova et al. (2011)
Methylocystis bryophila	S284	Methylocystaceae	Acetate	Belova et al. (2013)
Methylocella palustris	K	Beijerinckiaceae	Organic acids, alcohols	Dedysh et al. (2000)
Methylocella tundrae	T4	Beijerinckiaceae	Organic acids, alcohols	Dedysh et al. (2004)
Methylocystis heyeri	H2	Methylocystaceae	Acetate	Dedysh et al. (2007)
Methylocella silvestris	BL2	Beijerinckiaceae	Organic acids, alcohols, ethane, propane	Dunfield et al. (2003)
Methylocapsa aurea	KYG	Beijerinckiaceae	Acetate	Dunfield et al. (2010)
Methylocella tundrae	PC1	Beijerinckiaceae	Organic acids, alcohols, ethane, propane	Haque et al. (2019)

Methylocystis sp.	SB2	Methylocystaceae	Acetate, ethanol	Im et al. (2011)
Methylocystis	uncultivated	Methylocystaceae	Acetate	Leng et al. (2015)
Methylocystis hirsuta	CSC1	Methylocystaceae	Acetate	Lindner et al. (2007)
Methyloceanibacter methanicus	R-67174	Rhizobiales incertae sedis	Acetate	Vekeman et al. (2016)
Methylocella silvestris	TVC	Beijerinckiaceae	Organic acids, ethanol, propane	Wang et al. (2018)

As shown in the table, the preferential form of C substrate differs by facultative methanotroph species. Therefore, it can be inferred that the preferential form of C utilized by facultative methanotrophs may be determined by the methanotrophic community composition of soils.

References

- Belova, S. E. et al. Acetate utilization as a survival strategy of peat-inhabiting *Methylocystis* spp. *Environ. Microbiol. Rep.* **3**, 36–46 (2011).
- Belova, S. E., Kulichevskaya, I. S., Bodelier, P. L. E. & Dedysh, S. N. *Methylocystis bryophila* sp. nov., a facultatively methanotrophic bacterium from acidic Sphagnum peat, and emended description of the genus *Methylocystis* (ex Whittenbury et al. 1970) Bowman et al. 1993. *Int. J. Syst. Evol. Microbiol.* **63**, 1096–1104 (2013).
- Dedysh, S. N. et al. *Methylocystis heyeri* sp. nov., a novel type II methanotrophic bacterium possessing ‘signature’ fatty acids of type I methanotrophs. *Int. J. Syst. Evol. Microbiol.* **57**, 472–479 (2007).
- Dedysh, S. N. et al. *Methylocella tundrae* sp. nov., a novel methanotrophic bacterium

- from acidic tundra peatlands. *Int. J. Syst. Evol. Microbiol.* **54**, 151–156 (2004).
- Dedysh, S. N. et al. *Methylocella palustris* gen. nov., sp. nov., a new methane-oxidizing acidophilic bacterium from peat bogs, representing a novel subtype of serine-pathway methanotrophs Svetlana. *Int. J. Syst. Evol. Microbiol.* **50**, 955–969 (2002).
- Dunfield, P. F., Belova, S. E., Vorob'ev, A. V., Cornish, S. L. & Dedysh, S. N. *Methylocapsa aurea* sp. nov., a facultative methanotroph possessing a particulate methane monooxygenase, and emended description of the genus *Methylocapsa*. *Int. J. Syst. Evol. Microbiol.* **60**, 2659–2664 (2010).
- Dunfield, P. F., Khmelenina, V. N., Suzina, N. E., Trotsenko, Y. A. & Dedysh, S. N. *Methylocella silvestris* sp. nov., a novel methanotroph isolated from an acidic forest cambisol. *Int. J. Syst. Evol. Microbiol.* **53**, 1231–1239 (2003).
- Farhan Ul Haque, M., Crombie, A. T. & Murrell, J. C. Novel facultative *Methylocella* strains are active methane consumers at terrestrial natural gas seeps. *Microbiome* **7**, 1–17 (2019).
- Im, J., Lee, S. W., Yoon, S., Dispirito, A. A. & Semrau, J. D. Characterization of a novel facultative *Methylocystis* species capable of growth on methane, acetate and ethanol. *Environ. Microbiol. Rep.* **3**, 174–181 (2011).
- Leng, L., Chang, J., Geng, K., Lu, Y. & Ma, K. Uncultivated *Methylocystis* Species in Paddy Soil Include Facultative Methanotrophs that Utilize Acetate. *Microb. Ecol.* **70**, 88–96 (2015).
- Lindner, A. S. et al. *Methylocystis hirsuta* sp. nov., a novel methanotroph isolated from a groundwater aquifer. *Int. J. Syst. Evol. Microbiol.* **57**, 1891–1900 (2007).
- Vekeman, B. et al. New *Methyloceanibacter* diversity from North Sea sediments includes methanotroph containing solely the soluble methane monooxygenase. *Environ. Microbiol.* **18**, 4523–4536 (2016).
- Wang, J. et al. Draft Genome Sequence of *Methylocella silvestris* TVC, a Facultative Methanotroph Isolated from Permafrost. *Genome Announc.* **6**, e00040-18 (2018).

25. L84 – not fully accurate – Zhou et al. proposed that the inhibitory effect of ethylene released by drought stressed plants was a significant driver, not the effect of drought on methanotrophs.

Response: The reviewer made a valid point. As reported by Schnell et al. (1996), water stress can directly affect methanotrophs, leading to the inhibition of CH₄ oxidation. Therefore, we believe that low water availability can also act as a limiting factor for CH₄ oxidation by inducing stress on methanotrophs. The sentence has been revised as follows:

Line 102-106: “Under low soil moisture, water stress can promote ethylene production by plants, which inhibits CH₄ oxidation in soil^{30,31}. In addition, water stress on methanotrophs can result in reducing their activity²⁹. On the other hand, high soil moisture inhibits methanotrophic activity by decreasing gas diffusivity³⁰.”

26. L88-92 – can you discuss this in relation to the findings in *Science of the Total Environment* 757 (2021) 144089

Response: We added a discussion on the plant drought effect in line 103-105.

27. L99 – what defines highly saturated? Seems to be 1/6 of the time for temperate forests, so not uncommon.

Response: We defined a highly saturated period as 'when the measurement was taken immediately after a rainfall event or snow melt.' In temperate forest, measurement in August 2018 and July 2019 were conducted right after rain event and measurement in December 2018 and February 2019 were conducted right after snow melt due to warmer temperature than usual. In subtropical forest, measurement in May 2018 and December 2019 were performed right after rain event. As reviewers' concern, saturation for 2 months out of 12 months in temperate forest is not common. Thus, we define highly saturated period when the water-filled pore space is higher than 60% and 85% in temperate forest and subtropical forest, respectively (Fig. R12).

Fig. R12. Seasonal variation of water-filled pore space in subtropical forest (left) and temperate forest (right).

In brief, we define the data measured in December 2018 and February 2019 in temperate forest as the highly saturated period, while the data measured in May 2018 in the subtropical forest is also defined as the highly saturated period. We then performed regression analysis and found a bit weak but still significant positive relationship between CH₄ uptake rate and SWC as follows:

Fig. R13. Relationship between SWC and soil CH₄ uptake rate in subtropical forest (left) and temperate forest (right) after exclusion of highly saturated periods.

The definition of highly saturated period was added to the Method section as follows: Line 260-264: “Highly saturated periods were defined as periods when the measurements were taken right after a rain event or snowmelt, and when the water-filled pore space was higher than 85% and 60% in subtropical forests and temperate forests, respectively. Briefly, April 2018 in the subtropical forest and December 2018 and February 2019 in the temperate forest were defined as highly saturated periods.”

In addition, Supplementary Fig. 5 (Supplementary Fig. 3 in original manuscript) was revised as follows:

Supplementary Fig. 5. Relationship between soil CH₄ uptake rate and soil water content.

28. L105-107 – *how? Work with ethylene and drought stress may provide a pathway, but the work on this to date shows it is a 2nd order effect well below the potential for too much soil moisture to reduce gas diffusion into soil.*

Response: We believe that in terms of seasonal variation within a site, too much soil moisture induced by rain events or snow melt can significantly decrease methanotrophic activity. However, in terms of between-site variation, the mean CH₄ uptake rate is positively related to the soil moisture, as Gatica et al. (2020) reported a positive relationship between the mean CH₄ uptake rate and mean precipitation. This is further supported by our field measurements that the highest mean CH₄ uptake rate was observed in oak forest and moderate intensity plot where the highest soil moisture was observed in subtropical forest and temperate forest, respectively.

29. L122 – *pure-culture studies in the context of microbial diversity...? Some additional explanation is needed please.*

Response: We added sentences as follows:

Line 142-146: “For example, a culture study found the higher CH₄ oxidation rate for cultured methanotrophs with diverse heterotrophs compared to samples incubated with methanotrophs only³⁶. It was suggested that this mutual relationship was based on the fact that certain heterotrophs produce essential metabolites for methanotrophs, thus stimulating CH₄ oxidation activity.”

Reference

Ho, A. et al. The more, the merrier: Heterotroph richness stimulates methanotrophic activity. *ISME J.* **8**, 1945–1948 (2014).

30. L136 – *I am not a fan of Fig 2c, as described above.*

Response: Please see the response to comment 7.

31. L141 – *This sentence need to be rephrased. Also, the content has been mentioned earlier in the text.*

Response: The sentences have been revised as follows:

Line 161-165: “While SOC has been regarded to rather reduce the forest CH₄ sink by affecting methanogenesis^{32,34}, the relationship we found in the field observations and meta-analysis suggest that the positive effect of SOC on soil CH₄ uptake outweighs the negative effect at both regional and global scales. These results further suggest the need to incorporate the positive effect of SOC into the estimation model.”

Revised global forest CH₄ sink estimation model

32. *Figs 3a and 3b are not cited in the text.*

Response: Fig. 3a and 3b have been referred.

32. L160 – *define bottom up and top down estimation*

Response: Bottom-up estimates include process-based models for estimating land surface emissions and atmospheric chemistry, inventories of anthropogenic emissions, and data-driven extrapolations. Top-down estimates indicate atmospheric observations within an atmospheric inverse-modeling framework (Saunois et al., 2020).

We briefly define bottom-up and top-down estimation as follows:

Line 179-182: “As previous bottom-up estimation using a process-based model underestimated the global upland CH₄ sink by 8 Tg CH₄ yr⁻¹ in comparison with the top-down estimation that uses the atmospheric inverse-modeling framework⁴⁷, our revised model reconciles the disparity between the two different approaches.”

Reference

Saunois, M. *et al.* The Global Methane Budget 2000 – 2017. *Earth Syst. Sci. Data* **12**, 1561–1623 (2020).

33. L162-166 – *what were the significance values? Are you talking about changes in the amount of the uncertainty, or changes in the means values for the different biomes? I see changes in uncertainty, but not the mean.*

Response: The reviewer made a valid point. The sentences have been revised as follows:

Line 183-186: “Compared to TEM-DEF, TEM-SOC estimated 38% and 341% higher CH₄ sink in temperate forest (13.92 ± 1.44 Tg CH₄ yr⁻¹ versus 10.08 ± 3.28 Tg CH₄ yr⁻¹) and boreal forest (5.82 ± 1.27 Tg CH₄ yr⁻¹ versus 1.32 ± 0.44 Tg CH₄ yr⁻¹), respectively, whereas estimated 23% lower CH₄ sink in tropical forest (4.65 ± 0.37 Tg CH₄ yr⁻¹ versus 6.06 ± 2.8 Tg CH₄ yr⁻¹) (Fig. 3c).”

34. 168-170 – *this sentence needs to be rewritten for clarity.*

Response: The sentence has been revised as follows:

Line 188-195: In the regions characterized by predominantly low mean temperatures, such as boreal forests, it has been proposed that CH₄ and O₂ availability are not the primary limiting factors for CH₄ uptake^{32,33}. Rather, the biological activity of methanotrophs, which may be subject to physiological control by other environmental factors such as nutrient availability and pH, limits the process. Thus, in regions with low mean temperature, the role of soil variables that can promote the biological oxidation of CH₄ physiologically becomes more critical than CH₄ and O₂ availability.”

Methods

35. L322 – *The thinning intensity is a very shallow gradient, and I think that more extreme gradients would have created more useful outcomes. However, I assume you*

were using what was available to you at the time, and could not choose this. However, 300, 500, 800 and 1100 stems per ha would have been far more interesting to assess.

Response: The reviewer made a valid point that the intensity of thinning in this study was rather shallow. Although the % of trees removed in this study was a bit small, as reported in the manuscript, we could see significantly different soil characteristics including SOM content and SWC content among the different intensities. Therefore, we believe that the data obtained in the study site is valid to support our main finding, the positive relationship between SOM content and soil CH₄ uptake. We agree with the reviewer and a study on the response of CH₄ uptake to extreme thinning intensity is needed in the future to better understand the impact of thinning on forest CH₄ sink.

36. L350 – *with, not of*

Response: The word has been revised.

37. L359 – *“forest” as a search term covers many things... planted, natural, exotic, indigenous etc. I think you could gain some further value for your study by looking at different forest types as this would add another context to your results – if the number of replicates allows.*

Response: The reviewer made a valid point. We revisited the collected papers again and made an effort to identify additional categories that could yield interesting findings. There were difficulties in finding other categories since each of the papers missing different information. Finally, we categorized the data by the type of dominant trees in the forests (broadleaf vs needleleaf vs mixed forest) based on the previous study result that the type of leaf (broadleaf vs needleleaf) affects soil CH₄ oxidation due to the different chemical composition.

Fig. R14. Relationship between SOC content and CH₄ uptake rate in global forests with different types.

We found significant positive relationships between SOC content and CH₄ uptake rate across three distinct forest types worldwide. These results imply that SOC has a favorable impact on soil CH₄ uptake within each of these forest types.

In addition, Supplementary Fig. 8 shows the different magnitude of response of forest CH₄ uptake to SOC content in different latitudinal ranges. This figure supports one of the important findings of this study that the positive impact of SOC on forest CH₄ uptake is weaker in low latitudes.

38. L362 – *define the source of 1.72*

Response: The sentence has been revised as follows:

Line 283-286: “Collected SOM contents ($N = 5$) were converted to SOC content by using the simple equation: $SOM = SOC \times 1.72$ (van Bemmelen factor⁵⁴). The van Bemmelen factor has been widely used to convert SOM content to SOC content⁵⁵.”

References

Van Bemmelen, J. Über Die Bestimmung Des Wassers, Des Humus, Des Schwefels, Der in Den Colloïdalen Silikaten Gebundenen Kieselsäure, Des Mangans U. S. W. Im Ackerboden. *Die Landwirthschaftlichen Versuchs-Stationen* **37**, 279–290 (1890).

Heaton, L., Fullen, M. A. & Bhattacharyya, R. Critical Analysis of the van Bemmelen Conversion Factor used to Convert Soil Organic Matter Data to Soil Organic Carbon Data: Comparative Analyses in a UK Loamy Sand Soil. *Espaço Aberto* **6**, 35–44 (2016).

39. L364 – sometimes carbonate in forest soils is not negligible – e.g. *Applied Spectroscopy* 64(10):1167-75 and others

Response: In the study of Tatzber et al. (2010), 422 samples out of 504 samples were carbonate-free. Furthermore, the distribution of carbonate concentration of 82 carbonate-containing samples was concentrated in lower content. In general, soil inorganic carbon is usually found in arid or semi-arid regions (Lal & Kimble, 2000), whereas the majority of study sites incorporated in our meta-analysis were located outside of such regions (Fig. R15). Therefore, we believe that the majority of TC data we collected is comparable to SOC content.

Fig. R15. Map of the global arid region based on the Köppen-Geiger climate classification (Beck et al., 2018). The blue points show the location of forest sites used

for the meta-analysis. (BWh = Arid Desert (hot), BWk = Arid Desert (cold), BSh = Arid Steppe (hot), BSk = Arid steppe (cold))

References

Lal, R. & Kimble, J. M. *Global Climate Change And Pedogenic Carbonates*. CRC Press. 1–14 (2000).

Beck, H. E. et al. Present and future köppen-geiger climate classification maps at 1-km resolution. *Sci. Data* **5**, 1–12 (2018).

40. L408-415 – much more detail is required to communicate why the tests you used were chosen, and how you tested their inherent assumptions against the data you have to make sure they were the best choice.

Response: We added detailed explanations of statistical analysis as the reviewer suggests. Please see the response to comment 7.

Simeon Smail

Reviewer #2 (Remarks to the Author):

In this manuscript, the authors stressed the dominant role of soil organic carbon for CH₄ sinks in global forest soils, which is very sceptical.

Response: We thank the reviewer for taking the time to review our manuscript. We hope that the support for the manuscript voiced by the other reviewers as well as our response to reviewers' comments will provide reassurance as to the importance of our findings. The reviewers' comments were carefully addressed below.

We acknowledge that SOC is not the only variable that 'dominantly' controls CH₄ uptake in forests considering the complexity of forest soil CH₄ oxidation process. Nonetheless, it is our conviction that SOC represents a crucial factor in regulating the uptake of methane (CH₄) by forests, as we have observed. Despite this, the significance of SOC in this regard has not been fully recognized nor has it been adequately incorporated into current estimation models. Previous studies have proposed that SOC content exerts a negative influence on forest CH₄ uptake. This is attributed to the fact that SOC serves as a source of carbon substrates for methanogens, and increased carbon supply may stimulate heterotrophic metabolism, thereby outcompeting methanotrophy. However, this assumption does not consider the diverse functions of SOC on soil variables such as nutrient availability and pore space that are also associated with soil CH₄ oxidation. The novel aspect of our study is that the effect of SOC on forest CH₄ uptake is not negative but positive. This is supported by our field measurement in two sites: 1) forests with different tree species and 2) forests with the same species but different tree densities. In addition, the meta-analysis further supports the positive relationship between SOC content and forest CH₄ uptake rate across the global forests. Finally, the revised process-based model on global forest CH₄ sink suggests that incorporation of the effect of SOC reduces the uncertainty of the model and better represents the *in-situ* measurement.

1. Firstly, methane oxidizers get the majority of C from CH₄ being oxidized, not SOC.

Response: We agree with the reviewer that methanotrophs utilize CH₄ as their primary C source and other C compounds provided by SOC are not their main C source. However, the positive effect of other C compounds on soil CH₄ oxidation was reported in previous studies (Jensen et al., 1998; Sullivan et al., 2013). The authors suggested that in ecosystems where CH₄ concentration is low and high affinity methanotrophs play a key role in oxidizing CH₄, such as in forests, methanotrophs utilize other C compounds, thereby supporting the production and activity of methane monooxygenase enzymes. Our proposition is that the regulation of methane (CH₄) oxidation in SOC occurs through a range of multifaceted mechanisms. This assertion is based on the fact that SOC is a complex entity that encompasses a diverse array of materials, and it exerts various functions on soil characteristics, which contribute to the complexity of its influence on CH₄ oxidation in soil. In this manuscript, we have identified three potential mechanisms by which SOC can enhance forest soil CH₄ uptake: 1) by alleviating water stress, 2) by providing alternative C substrates which subsidize the production of methane monooxygenase enzymes, and 3) by improving pore space of soil. We would like to point out that the influence of the alternative C compound is one of the mechanisms that we propose, not the 'main' and single mechanism. Rather than a single mechanism dominating the positive effect of SOC, we believe that SOC positively affects CH₄ uptake through the combination of diverse mechanisms.

References

- Jensen, S., Priemé, A. & Bakken, L. Methanol improves methane uptake in starved methanotrophic microorganisms. *Appl. Environ. Microbiol.* **64**, 1143–1146 (1998).
- Sullivan, B. W., Selmants, P. C. & Hart, S. C. Does dissolved organic carbon regulate biological methane oxidation in semiarid soils? *Glob. Chang. Biol.* **19**, 2149–2157 (2013).

2. Secondly, the main results are inconsistent. The SOC-Model resulted in an increase in CH₄ sink for the temperate forests but a decrease in the subtropical forest, which was inconsistent with the observation as DOC showed a much higher positive correlation with CH₄ uptake in the subtropical forest compared with that in the temperate forest.

Response: In our model, function of SOC represents a positive correlation between SOC and CH₄ uptake (Supplementary Fig. 9), and the same function was used to estimate CH₄ sink across all forest types. Therefore, the decrease in CH₄ sink in tropical forests in our model does not indicate that the relationship between SOC and CH₄ sink is negative in this region. Instead, tropical forests exhibited a slight decrease in forest soil sink (statistically not significant), most likely because the positive effect of SOC is weak in low latitude regions as shown in Supplementary Fig. 8. In addition, the relatively low SOC content of tropical forests may induce a slight decrease in CH₄ sink when the positive influence of SOC is incorporated into the model. Although the subtropical forest site is located in the subtropical region, the climate of this site is closer to the temperate region rather than the tropical region, with a mean annual temperature of 17°C. This may explain the strong positive correlation between DOC content and CH₄ uptake in the subtropical forest site. As mentioned above, the alternative C compound provided by SOC is one of the mechanisms that we propose, but it's not the only dominant mechanism. Therefore, the strong positive correlation between DOC content and CH₄ uptake rate observed in the subtropical forest does not necessarily imply that the role of SOC in this region is more prominent than in the temperate forest.

3. In lines 142-145, which data (observations and metadata) could suggest that the positive effect of SOC on soil CH₄ uptake is larger than the negative effect on both regional and global scales? How?

Response: If the negative effect of SOC on soil CH₄ uptake is larger than the positive effect, negative relationship between the two variables should have appeared. However, we found significant positive relationship between those two variables both

in field observation and meta-analysis as follows:

Fig. 1. Relationships between forest soil CH₄ uptake rate and SOM content in (a) subtropical forest with different tree species (*N* = 72) and (b) temperate coniferous forest with different thinning intensities (*N* = 258).

Fig. 2b. Relationship between forest CH₄ uptake rate and SOC content based on the selected sites around the globe ($N = 204$).

Based on the *in-situ* measurement, we concluded that the relationship between SOM content and forest CH₄ uptake rate is positively related on a regional scale. A positive correlation observed in subtropical forest with different tree species further suggests that SOM plays a role in explaining the variability of soil CH₄ uptake between species on a regional scale. A positive correlation observed in temperate forest with different thinning intensities suggests that SOM content is one of the controlling variables that could differentiate soil CH₄ uptake rate between forests with the same dominant tree species but differing environmental conditions such as tree density and soil characteristics. This is supported by the positive correlation between SOM content and soil CH₄ uptake rate observed in the temperate forest, where the dominant species is *Pinus Koraiensis* but the tree density and soil characteristics vary significantly between sites. As depicted in Figure 2, a total of 81 published papers were gathered and 204 data sets were extracted through a systematic review process. Based on the global meta-analysis, we concluded that the relationship between the two variables is still positive on a global scale.

4. In lines 166-172, previous studies have reported high methane uptake rates with low temperatures (even below zero) and low temperatures limited methane uptake mainly by freezing soil, thus reducing the availability of soil water and O₂. How could high SOC promote CH₄ uptake in this situation? The microbiological data evidence is required to support this point.

Response: The term 'low temperature' in the sentences was used to refer to regions with low annual average temperatures, such as boreal forests. We recognize that readers may misconstrue the term 'low temperature' to imply excessively low temperatures that cause soil freezing. Therefore, the sentence has been revised as follows:

Line 188-195: “In the regions characterized by predominantly low mean temperatures, such as boreal forests, it has been proposed that CH₄ and O₂ availability are not the primary limiting factors for CH₄ uptake^{32,33}. Rather, the biological activity of methanotrophs, which may be subject to physiological control by other environmental factors such as nutrient availability and pH, limits the process. Thus, in regions with low mean temperature, the role of soil variables that can promote the biological oxidation of CH₄ physiologically becomes more critical than CH₄ and O₂ availability.”

In the regions with low mean annual temperature including boreal forest, CH₄ and O₂ availability is not the limiting factor since O₂ diffusion potential from the atmosphere to soil is higher than soil O₂ consumption due to low microbial activity (Nedwell & Watson, 1995). In these regions, however, other environmental variables that can physiologically control the activity of methanotrophs such as nutrient availability, temperature, pH, and water availability are the limiting factor and can control methanotrophic activity (Tang et al., 2010; Gatica et al., 2021). That being said, the role of SOC, which can control diverse soil functioning including nutrient availability and water availability, is larger in a colder region. This is further supported by our meta-analysis that the slope of the regression between SOC content and CH₄ uptake rate is steeper in the high latitude (40°-70° N&S) than in the mid latitude (20°-40° N&S) and low latitude (0°-20° N&S).

Supplementary Fig. 8. Relationship between soil CH₄ uptake rate and soil organic carbon content in different latitudinal ranges (Supplementary Fig. 7 in original manuscript).

In addition, we categorized data by mean annual temperature (0-10 °C, 10-20 °C, 20-30 °C) and performed regression analysis between SOC content and CH₄ uptake rate. We found that the forests in the region with low mean annual temperature (0-10 °C) exhibited the steepest regression slope whereas the forests in the region with high temperature (20-30 °C) exhibited the least regression slope (Fig. R16). This result further reinforces our finding that the role of SOC on CH₄ uptake is more important in the region with low mean annual temperature.

Fig. R16. Relationship between soil CH₄ uptake rate and soil organic carbon content in different mean annual temperature ranges.

When the soil is frozen and material transportation via water stops, the influence of the proposed mechanisms in which SOC enhances the CH₄ uptake, such as through improving water availability or alternative C availability, may become weakened. Therefore, we believe that the role of SOC on CH₄ uptake is particularly important

when the soils are not frozen. In fact, the majority of CH₄ sink is observed in when the soils are thawed whereas the amount of CH₄ sink is relatively low when the soils are frozen in boreal forest most likely due to the snow cover and frozen soil (Sundqvist et al., 2015; Zona et al., 2016; Helbig et al., 2017), indicating that the CH₄ sink in frozen season is less important. Meanwhile, Jensen et al. (1998) suggested that soil labile carbon stimulates methanotrophic growth by providing a substitute energy source to methanotrophs when no consumable CH₄ is available. The findings of this study indicate that in situations where CH₄ is scarce due to freezing, SOC can positively affect methanotrophs by providing alternative C sources to sustain their activities.

References

- Fang, H. J. *et al.* Effects of multiple environmental factors on CO₂ emission and CH₄ uptake from old-growth forest soils. *Biogeosciences* **7**, 395–407 (2010).
- Gatica, G., Fernández, M. E., Juliarena, M. P. & Gyenge, J. Environmental and anthropogenic drivers of soil methane fluxes in forests: Global patterns and among-biomes differences. *Glob. Chang. Biol.* **26**, 6604–6615 (2020).
- Helbig, M., Quinton, W. L. & Sonntag, O. Warmer spring conditions increase annual methane emissions from a boreal peat landscape with sporadic permafrost. *Environ. Res. Lett.* **12**, (2017).
- Jensen, S., Priemé, A. & Bakken, L. Methanol improves methane uptake in starved methanotrophic microorganisms. *Appl. Environ. Microbiol.* **64**, 1143–1146 (1998).
- Nedwell, D. B. & Watson, A. CH₄ production, oxidation and emission in a U.K. ombrotrophic peat bog: Influence of SO₄²⁻ from acid rain. *Soil Biol. Biochem.* **27**, 893–903 (1995).
- Sundqvist, E., Mölder, M., Crill, P., Kljun, N. & Lindroth, A. Methane exchange in a boreal forest estimated by gradient method. *Tellus, Ser. B Chem. Phys. Meteorol.* **67**, (2015).
- Zona, D. et al. Cold season emissions dominate the Arctic tundra methane budget.

Proc. Natl. Acad. Sci. U. S. A. **113**, 40–45 (2016).

5. High heterotrophs activities in SOC-rich soil may benefit methanotroph growth and activity, but it should not be the key determinant.

Response: As the reviewer suggests, the positive influence of SOC on CH₄ uptake through improving the diversity of heterotrophic microbes is one of the mechanisms we suggest, but not the key mechanisms. We included L138-151 to suggest additional possible mechanisms that SOC positively influences CH₄ sink in forests. Those mechanisms discussed in this paragraph are based on the previous literature, and we do not indicate these mechanisms are the key determinant.

Reviewer #3 (Remarks to the Author):

This manuscript reports experiments and a global analysis of published data to suggest that the ability of forest soil to be a methane sink is positively correlated with SOC content. They suggest this is largely due to greater soil wetting with higher SOC but may also be due to higher microbial diversity, direct interaction with organic matter such as acetate and methanol, or increase methanogenesis. I find this paper important, well-written, and compelling. My only comment is that the section starting at line 168 has a long discussion of high and low temperatures. It would really help to know what values are considered to be high and low in this context.

Response: We thank the reviewer for taking time to review our manuscript and providing a positive comment.

The term 'low temperature' in the sentences was used to refer to regions with low annual average temperatures such as boreal forests. We acknowledge that the term 'low temperature' may be misinterpreted by readers to mean temperatures that are so low that soil freezes over. Therefore, the sentence has been revised as follows:

Line 188-195: "In the regions characterized by predominantly low mean temperatures, such as boreal forests, it has been proposed that CH₄ and O₂ availability are not the primary limiting factors for CH₄ uptake^{32,33}. Rather, the biological activity of methanotrophs, which may be subject to physiological control by other environmental factors such as nutrient availability and pH, limits the process. Thus, in regions with low mean temperature, the role of soil variables that can promote the biological oxidation of CH₄ physiologically becomes more critical than CH₄ and O₂ availability."

In the regions with low mean annual temperature including boreal forest, CH₄ and O₂ availability is not the limiting factor since O₂ diffusion potential from the atmosphere to soil is higher than soil O₂ consumption due to low microbial activity (Watson, 1995). In

these regions, however, other environmental variables that can physiologically control the activity of methanotrophs such as nutrient availability, pH, and water availability are the primary limiting factor and can control methanotrophic activity. As shown in Supplementary Fig. 8, the slope of the regression between CH₄ uptake and SOC content gets steeper with a high latitudinal range. This result indicates that the role of SOC, which can physiologically control methanotrophic activity, becomes more important in colder regions. Although we could not able to defines what values are considered to be high and low temperate, our data suggest that the role of SOC becomes greater in the regions with low mean temperatures.

Supplementary Fig. 8. Relationship between soil CH₄ uptake rate and soil organic carbon content in different latitudinal ranges (Supplementary Fig. 7 in original manuscript).

In addition, we categorized data by mean annual temperature (0-10°C, 10-20°C, 20-30°C) and performed regression analysis between SOC content and CH₄ uptake rate. We found that the forests in the region with low mean annual temperature (0-10°C) exhibited the steepest regression slope whereas the forests in the region with high temperature (20-30°C) exhibited the least regression slope (Fig. R16). This result provides additional support for the conclusion that the significance of SOC in CH₄

uptake is greater in regions with low mean annual temperatures.

Fig. R16. Relationship between soil CH₄ uptake rate and soil organic carbon content in different mean annual temperature ranges.

Reviewer #1 (Remarks to the Author):

Thanks you for your detailed responses to my questions. I am happy with your reasoning and the alterations you have made to the text, where needed.